# GroEL actively stimulates folding of the endogenous substrate protein PepQ

Jeremy Weaver[1,*,†], Mengqiu Jiang[1,2,*], Andrew Roth[1], Jason Puchalla[3], Junjie Zhang[1] & Hays S. Rye[1]

Many essential proteins cannot fold without help from chaperonins, like the GroELS system of *Escherichia coli*. How chaperonins accelerate protein folding remains controversial. Here we test key predictions of both passive and active models of GroELS-stimulated folding, using the endogenous *E. coli* metalloprotease PepQ. While GroELS increases the folding rate of PepQ by over 15-fold, we demonstrate that slow spontaneous folding of PepQ is not caused by aggregation. Fluorescence measurements suggest that, when folding inside the GroEL-GroES cavity, PepQ populates conformations not observed during spontaneous folding in free solution. Using cryo-electron microscopy, we show that the GroEL C-termini make physical contact with the PepQ folding intermediate and help retain it deep within the GroEL cavity, resulting in reduced compactness of the PepQ monomer. Our findings strongly support an active model of chaperonin-mediated protein folding, where partial unfolding of misfolded intermediates plays a key role.

[1] Department of Biochemistry and Biophysics, Texas A&M University, College Station, Texas 77845, USA. [2] State Key Laboratory of Biocontrol, School of Life Science, Sun Yat-sen University, Guangzhou, Guangdong 510275, China. [3] Department of Physics, Princeton University, Princeton, New Jersey 08544, USA. * These authors contributed equally to this work. † Present address: Division of Molecular and Cellular Biology, NICHD, National Institutes of Health, Bethesda, Maryland 20892, USA. Correspondence and requests for materials should be addressed to J.Z. (email: junjiez@tamu.edu) or to H.S.R. (email: haysrye@tamu.edu).

Folding is a highly error prone process for many large and essential cellular proteins. Misfolding and aggregation often overwhelm the delicate thermodynamic balance that drives a protein toward its native state. Throughout evolutionary history, living systems have solved this problem with specialized, ATP-powered machines known as molecular chaperones[1]. The Hsp60s or chaperonins are a central and essential family of the molecular chaperones, and the GroELS chaperonin system of *Escherichia coli* is one of the best studied examples[2–5]. GroEL is a homo-oligomer of 14, 57 kDa subunits, that is arranged in two, seven membered rings stacked back-to-back. Each ring contains a large, open, solvent-filled cavity[6]. The inner cavity surface of the uppermost domain (the apical domain) is lined with hydrophobic amino acids that capture non-native substrate proteins[7,8]. Substrate proteins that strictly depend upon GroEL for folding (so-called stringent substrate proteins) must be briefly enclosed within a complex formed by a GroEL ring and the smaller, ring-shaped co-chaperonin GroES[9–12]. Formation of the GroEL-GroES complex first requires that a GroEL ring bind ATP, which triggers a series of conformational rearrangements of the GroEL ring, permitting GroES to bind and resulting in the encapsulation of the substrate protein. Enclosure of the substrate protein beneath GroES results in ejection and confinement of the protein inside the enlarged GroEL-GroES chamber (a *cis* complex) and initiation of protein folding. Folding continues within the isolated GroEL-GroES cavity for a brief period, until the complex is disassembled and the substrate protein, folded or not, is released back into free solution[9–11,13,14].

Despite this detailed structural and functional knowledge, current models of GroEL-assisted folding remain divided into two general types based upon whether GroEL is presumed to act passively or actively[2,3,5,15]. Passive models, like the Anfinsen cage or infinite dilution model, postulate that protein folding is only enhanced by GroELS because folding intermediates are prevented from aggregating by isolating them within the protective environment of the GroELS chamber[2,15]. Purely passive models implicitly assume that the folding of GroEL-dependent proteins are constrained only by the aggregation propensity of on-pathway folding intermediates. Active GroEL folding models, by contrast, assume that stringent GroEL-substrate proteins can and do populate off-pathway, kinetically trapped states. In this view, GroELS stimulates protein folding because these kinetically trapped intermediates benefit not only from protection against aggregation but also from additional, and essential, corrective actions provided by the chaperonin[3,16]. The mechanism of this corrective action remains controversial, but has been suggested to come from either (1) repetitive unfolding and iterative annealing[17,18] or (2) smoothing of a substrate protein's free energy landscape as a result of confinement inside the GroEL-GroES cavity, where either steric constraints and/or interactions within the chamber prevent unproductive folding pathways in favour of productive ones[3,15,16].

Several stringent substrate proteins have been shown to display folding behaviour that is consistent with one or more predictions of active GroEL folding models[19–21]. Some of the most detailed analysis to date has been conducted with ribulose-1,5-bisphosphate carboxylase oxygenase (RuBisCO) from *R. rubrum* and a double mutant of *E. coli* maltose binding protein (MBP)[19–24]. While highly suggestive, these studies nonetheless leave the importance of active folding unclear. General conclusions about the impact of active folding cannot be robustly drawn from such a small number of examples. In addition, in the case of RuBisCO, the mismatch between the biological source of the substrate protein (*Rhodospirillum rubrum*) and the chaperonin (*E. coli*), leaves the biological consequences of these findings open to interpretation. Similarly, in the case MBP, it was necessary to employ an engineered double-mutant of this protein in order to study GroEL-stimulated folding, because wild-type MBP neither interacts with, nor needs the chaperonin for folding in its natural biological context. Thus, a convincing demonstration of active folding assistance by GroEL of a stringent, endogenous *E. coli* substrate protein has remained elusive. A recent study on the assisted folding of the *E. coli* HTP synthase/lyase DapA sought to address this problem[25]. The results of this work suggested that DapA requires an active GroEL folding mechanism. However, a more recent study of DapA folding called key elements of this work into question[26].

In order to test the central predictions of passive and active models of chaperonin-mediated folding, we have re-examined the mechanism of GroELS-assisted protein folding using the biologically relevant, endogenous *E. coli* prolidase enzyme, PepQ. PepQ catalyses the hydrolysis of dipeptides that contain C-terminal proline residues[27,28]. It forms a homodimer, with each monomer (~50 kDa) built from two domains: a small, mixed α/β N-terminal domain and a pita-bread fold[29,30] C-terminal domain that contains the active site (Fig. 1a; ref. 27). Two independent proteomics studies predicted that PepQ requires the assistance of GroEL-GroES for folding *in vivo*[31,32]. In addition, PepQ is a member of a protein structural family that is not represented among the well-characterized GroEL-substrate proteins. Here, using a combination of enzymatic assays, single-molecule fluorescence techniques, and cryo-electron microscopy (EM), we demonstrate that GroEL actively alters the folding of PepQ. Initial capture of a kinetically trapped PepQ monomer by a GroEL ring results in substantial unfolding, a process that relies in part on a direct, physical interaction between the PepQ folding intermediate and the unstructured GroEL C-terminal tails. Subsequent encapsulation of the partially unfolded folding PepQ monomer within the GroEL/ES chamber fundamentally alters the folding trajectory of the protein, resulting in a faster and more efficient search for the native state.

## Results

**Slow spontaneous PepQ folding is not caused by aggregation.** Upon dilution from chemical denaturant, PepQ folds spontaneously at room temperature (23 °C) to a final yield of 50–60% with an observed half-time of ~20 min (Fig. 1b). However, in the presence of the cycling GroEL-GroES system, PepQ folds with an observed half-time of ~1 min to a final yield of 80–90% (Fig. 1b). Encapsulation of PepQ within a non-cycling chaperonin complex, composed of the GroEL single-ring mutant SR1 and GroES, also results in accelerated refolding, consistent with previous observations from other GroEL-substrate proteins (Fig. 1b; refs 9,25,33,34). Inside the static SR1-ES cavity, PepQ folds at a rate similar to that observed with cycling wild type GroEL at 23 °C, although it displays a consistently lower yield. Thus, while PepQ does not require GroEL to fold, the chaperonin accelerates the folding rate of the enzyme by 15–20-fold, while increasing the native state yield by ~40%.

The observation of slow spontaneous folding, in combination with a decreased native state yield, suggests that PepQ folding is inhibited by non-productive side reactions like misfolding or aggregation. We therefore examined the fate of the PepQ that fails to reach the native state. While this material remains fully in solution, over time it loses the ability to fold productively, even with assistance from GroEL (Supplementary Fig. 1). Approximately half of the PepQ population becomes refractory to GroEL-mediated folding with a time constant that is similar to that observed for productive spontaneous folding. Because the non-native states of many chaperonin-dependent proteins are highly prone to aggregation, we sought to determine whether

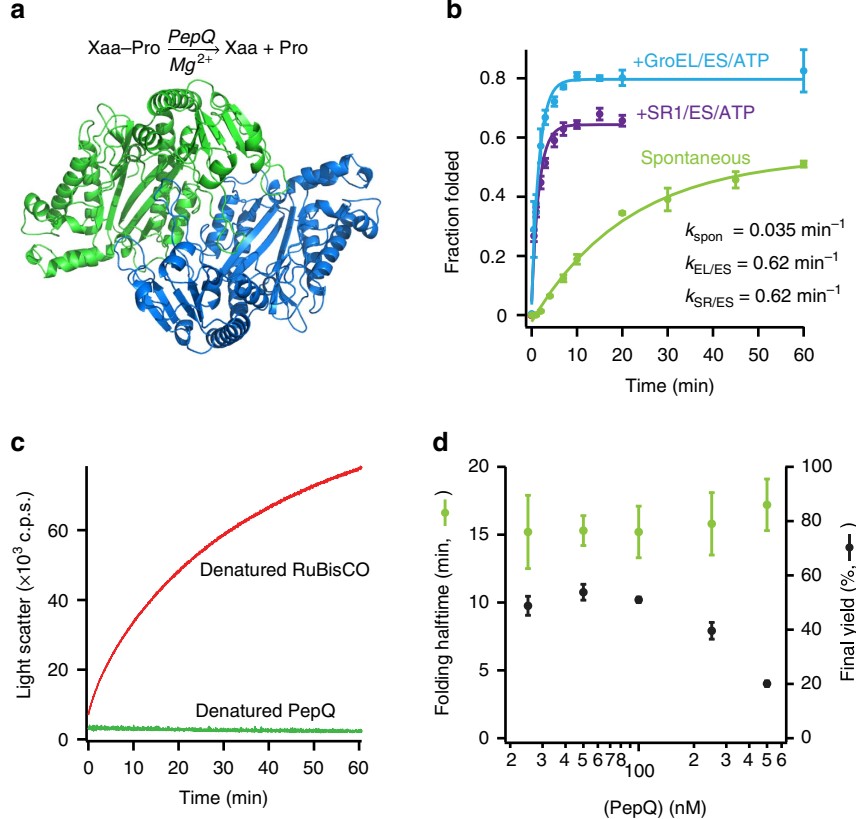

**Figure 1 | Stimulated folding of PepQ by GroEL does not depend on large-scale suppression of aggregation. (a)** The *E. coli* metalloprotease PepQ catalyses the hydrolysis of dipeptides containing C-terminal proline residues. The structure of the native PepQ homodimer (PDB ID: 4QR8) is shown, illustrating the pita-bread fold common to this enzyme family. **(b)** Refolding of PepQ was monitored by the recovery of enzymatic activity. PepQ was denatured in acid-urea and then diluted into either buffer alone (100 nM; spontaneous, green) or buffer containing GroEL (200 nM). The GroEL-PepQ binary complex was then supplemented with GroES (400 nM) and ATP (2 mM) to initiate folding (+GroEL/ES/ATP, blue). In a parallel experiment, denatured PepQ was bound to the single-ring mutant of GroEL, SR1 (300 nM), and refolded in the presence of GroES (600 nM) and ATP (2 mM; +SR1/ES/ATP, purple). Data were fit to a single-exponential rate law (solid lines), resulting in observed folding rate constants of $0.62 \pm 0.05$ min$^{-1}$ for GroEL, $0.62 \pm 0.09$ min$^{-1}$ for SR1 and $0.035 \pm 0.005$ min$^{-1}$ for the spontaneous reaction. Error bars show the standard deviation of three independent experiments. **(c)** Large-scale aggregation of PepQ and RuBisCO was examined by static light scattering at 340 nm. PepQ (green) and RuBisCO (red) were each denatured in acid-urea and then separately diluted into buffer at 23 °C (100 nM final monomer). Each trace is the average of three separate experiments. **(d)** The rate and yield of spontaneous PepQ folding as a function of enzyme concentration is shown. Chemically denatured, wild type PepQ was diluted 50-fold into buffer to yield spontaneous folding reactions at the indicated final monomer concentration. The folding rate at each protein concentration (green) and native state yield (black) are shown. Error bars show the s.d. of three independent folding experiments.

inefficient PepQ folding was due to aggregation. We first examined the static light scattering of a spontaneous folding reaction in which PepQ was rapidly diluted from denaturant into refolding buffer. Surprisingly, PepQ displayed no significant increase in light scattering, even after 1 h of incubation at 23 °C (Fig. 1c). By contrast, *R. rubrum* RuBisCO, a stringent GroEL-substrate protein well known to aggregate at 23 °C (refs 19,35,36) showed a rapid and substantial increase in light scattering under the same conditions (Fig. 1c). These observations indicate that denatured PepQ does not form high concentrations of large aggregates, at least under the conditions of the spontaneous folding assay. However, PepQ could form inhibitory aggregates that are too small or rare to be well detected by light scattering. If true, the observed rate of spontaneous PepQ folding should be a sensitive function of the total protein concentration. Strikingly, over a concentration range from 25 to 500 nM, the observed half-time of PepQ folding remained unchanged, although we did observe a decrease in the native state yield as the protein concentration was increased above 250 nM (Fig. 1d).

The concentration independence of the PepQ folding rate suggested that the slowness of spontaneous folding is not caused

by inhibitory aggregation. To further test this conclusion, we examined PepQ folding at low protein concentrations using a set of fluorescence-based assays. We first introduced a surface-exposed Cys residue into the first helix of the PepQ N-terminal domain (A24C), which permitted unique attachment of exogenous fluorescent probes (Supplementary Fig. 2A). Importantly, PepQ labelled at position 24 with small dyes like IAEDANS (PepQ-24ED), fluorescein (PepQ-24F), Oregon Green (PepQ-24OG) or tetramethyl rhodamine (PepQ-24TMR) displayed no apparent alteration in enzymatic activity or stability. Spontaneous folding of the PepQ-24ED variant displayed no significant difference compared to wild type PepQ, and the PepQ-24F, PepQ-24OG, PepQ-24TMR variants folded only slightly more slowly (~30%; Supplementary Fig. 2B). We used these labelled PepQ variants in an intermolecular Förster Resonance Energy Transfer (FRET) assay designed to examine aggregate formation during spontaneous PepQ folding[27]. In this assay, two differently labelled PepQ monomers were employed: PepQ-24ED as the donor and PepQ-24F as the acceptor. In the native PepQ dimer, these sites are positioned too far apart for Förster coupling (Supplementary Fig. 2A), so that any observed

FRET signal should report primarily on aggregate formation. When the two PepQ samples were mixed, denatured and diluted together into refolding buffer at 50 °C, formation of PepQ aggregates was readily observed as a robust FRET signal (80% FRET efficiency; Supplementary Fig. 2C). Surprisingly, when the same experiment was conducted under spontaneous folding conditions at 23 °C, the observed FRET efficiency was less than 4%, suggesting a lack of significant aggregation (Supplementary Fig. 2C).

We next examined PepQ folding and aggregation at extremely low protein concentrations using single-molecule detection techniques. First, samples of PepQ-24TMR were denatured in acid-urea and spontaneous folding was initiated by rapid dilution (50-fold) into refolding buffer at 23 °C, yielding a final monomer concentration of 2 nM. This sample was allowed to fold spontaneously at 23 °C and samples were removed and mixed with a large excess of GroEL at different time points. Excess GroEL was added to both quench the folding reaction and increase the effective diffusion time of uncommitted PepQ monomers, which were bound by the much larger GroEL tetradecamer[22,25]. Fluorescence correlation spectra (FCS) were then acquired for each time point and the fraction of folded versus non-native PepQ was extracted from each autocorrelation curve by comparison with two reference states: non-native PepQ-24TMR bound to GroEL and native PepQ-24TMR. The normalized autocorrelation curves of these two reference states are shown in Fig. 2a. The rate of spontaneous PepQ-24TMR folding, measured at 2 nM by FCS, closely recapitulates the rate of folding of the protein observed at 100 nM (Fig. 2b). More importantly, when the same experiment was conducted with fully cycling GroEL-GroES, folding of PepQ-24TMR was stimulated by the same 15–20-fold observed at higher concentrations (Fig. 2b).

Using single-molecule, two-colour co-incidence detection we next probed the assembly status of PepQ during spontaneous folding at 2 nM. As a control, we first examined formation of the native PepQ dimer. A 1:1 mixture of PepQ-24OG and PepQ-24TMR was denatured and refolded at a total PepQ concentration of 100 nM in the presence of the active GroEL-GroES system, in order to permit formation of PepQ dimers carrying both probes. This sample was then diluted to 100 pM PepQ and fluorescence bursts were collected using using a two-channel, confocal-type single-molecule microscope (Fig. 3a, inset). The native PepQ dimer was readily detectable as a robust fraction of coincident events (Fig. 3a). Notably, the observed coincident fraction (∼10%) was lower than the theoretically expected value of ∼50% for a 1:1 mixture of PepQ-24OG and PepQ-24TMR. This difference is most likely due to the much greater tendency of OG to convert to a long-lived dark (triplet) state, relative to TMR (Supplementary Fig. 2D–F), which results in a substantial decrease in observed co-incidence.

To examine PepQ monomer assembly during spontaneous folding, samples of PepQ-24OG and PepQ-24TMR were mixed at 1:1, denatured in acid-urea and then rapidly diluted (50-fold) into refolding buffer at 23 °C to initiate spontaneous folding at a final protein concentration of 2 nM monomer. This sample was incubated at 23 °C for 10 min, then diluted another 20-fold to a final PepQ monomer concentration of 100 pM. The native PepQ dimer does not readily form at a monomer concentration of 2 nM. However, it is possible that low-order, non-native aggregates stabilized by much larger contact surfaces might still form[36]. We therefore anticipated that any co-incidence observed between the two labelled PepQ monomers would have to result from such low-order aggregates. Importantly, the observed co-incidence was less than 1% (Fig. 3b). Even taking into account the reduced sensitivity caused by the differences in triplet state conversion of the OG and TMR dyes, these measurements

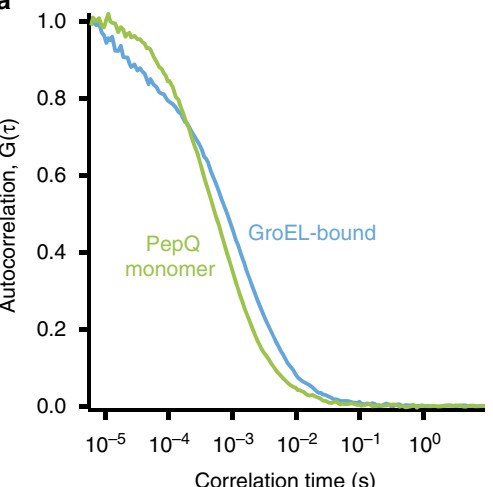

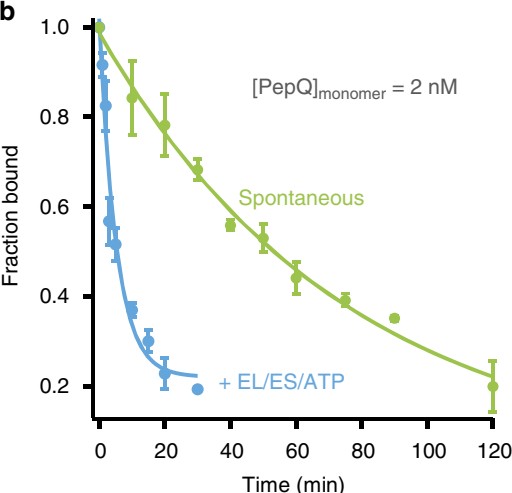

**Figure 2 | GroEL accelerates folding of PepQ at very low protein concentrations. (a)** The difference in diffusion time of the PepQ monomer in free solution, versus bound to a GroEL tetradecamer, can be detected by FCS. The observed FCS curves of the PepQ-24TMR monomer (2 nM), either alone (green) or bound to GroEL (1 μM; blue) are shown. **(b)** Refolding of PepQ-24TMR was monitored by FCS, using the observed shift in diffusion time shown in **a**. PepQ-24TMR was denatured in acid-urea and diluted either directly into buffer (2 nM; spontaneous, green) or into buffer containing wild type GroEL (1 μM). Refolding with GroEL was initiated by addition of GroES (2 μM) and ATP (2 mM; +EL/ES/ATP, blue). At the indicated times, GroEL-mediated folding was quenched by depletion of ATP before FCS measurement, while samples of the spontaneous reaction were mixed with GroEL alone (1 μM) before FCS measurement in order to quench folding and shift the diffusion time of any uncommitted PepQ monomer. The observed fractional change in diffusion time was fit to a single-exponential rate law (solid lines), resulting in rate constants of $0.19 \pm 0.04$ min$^{-1}$ for GroEL-mediated folding and $0.013 \pm 0.002$ min$^{-1}$ for spontaneous folding. Error bars show the s.d. of three experimental replicates.

indicate that, at most, 4–5% of the PepQ monomers could be found in an assembled state of any kind, including the smallest possible aggregates (non-native dimers), during spontaneous folding at 2 nM. In total, these observations demonstrate that slow spontaneous folding of PepQ cannot be due to inhibitory aggregation, but instead must result from the inherently inefficient conformational search of the PepQ monomer. In

addition, our data suggest that the PepQ monomers that do not reach the native state during spontaneous folding at low protein concentrations likely persist as kinetically trapped monomers.

**GroEL alters the folding trajectory of the PepQ monomer.** To achieve the large folding stimulation observed with PepQ, in the absence of aggregation, GroEL must actively alter how the protein folds. To investigate the nature of this alteration, we exploited the intrinsic tryptophan fluorescence of PepQ. Importantly, PepQ has multiple tryptophan residues, while GroEL and GroES are devoid of this amino acid. During spontaneous folding, the tryptophan fluorescence of PepQ displays a single, downward transition with a time constant of ∼125 s (Fig. 4a). The rate of this fluorescence decrease is substantially faster than the limiting rate at which PepQ spontaneously commits to the native state (Fig. 1b). This suggest that, at least for spontaneous folding, the observed shifts in tryptophan fluorescence report on transitions that precede the committed step of PepQ folding. By contrast, assisted folding of PepQ with the cycling GroEL-GroES system results in a rapid, early increase in tryptophan fluorescence ($\tau = \sim13$ s), which is followed by a subsequent decrease in fluorescence with a time constant of ∼73 s (Fig. 4b). The large increase in fluorescence observed with GroEL most likely reports on an early folding transition that occurs after the PepQ folding intermediate has been released into the GroEL-GroES cavity. It is unlikely that the early fluorescence rise is due to either GroES binding and encapsulation alone, or to simple release of the PepQ monomer into the cavity, as these events occur much faster than the observed rate of the PepQ fluorescence change[9,19,37]. Although PepQ folding with the cycling GroEL-GroES system rapidly becomes asynchronous, the transition between the increasing and decreasing fluorescence phases occurs after roughly one cavity lifetime at 23 °C (refs 22,37,38). This observation supports the idea that the increase in fluorescence occurs inside the GroEL-ES cavity. To directly test this conclusion, we employed SR1 to examine a single round of PepQ encapsulation and folding inside the GroEL-GroES cavity. Notably, PepQ confined within the SR1-GroES cavity also displays a rapid increase in fluorescence, but no subsequent decrease (Fig. 4c), confirming the conclusion that the early increase in PepQ fluorescence occurs within the GroEL-GroES cavity.

To further define how the folding behaviour of PepQ is altered by GroEL, we examined the impact of the GroEL C-termini on PepQ folding. We previously showed that a tailless GroEL variant (Δ526 GroEL) has a significantly reduced ability to assist the folding of the classically stringent GroEL-substrate protein, RuBisCO from *R. rubrum*[39]. Interestingly, removal of the C-termini has an even more pronounced negative impact on PepQ folding. Deletion of the C-termini from a cycling GroEL tetradecamer (Δ526) causes a nearly sixfold reduction in the observed PepQ folding rate (Fig. 5a), versus an approximate twofold reduction with RuBisCO (ref. 39). By contrast, removal of the C-termini from the non-cycling SR1 GroEL variant (SRΔ526) results in a more modest twofold decrease in the PepQ folding rate (Fig. 5a). Because deletion of the C-termini can, in some cases, result in premature substrate protein release before GroES binding and encapsulation[40], we considered whether the observed drop in PepQ folding rate with the Δ526 variants is simply due to a trivial decrease in encapsulation efficiency. However, Δ526 displays no substantial premature release of PepQ relative to full-length GroEL and the early escape of PepQ from SRΔ526, compared to SR1, is no greater than 10% (Supplementary Fig. 3). While consistent with previous observations with RuBisCO (ref. 40), this minor drop in encapsulation efficiency is too small to explain the reduction in observed folding rate.

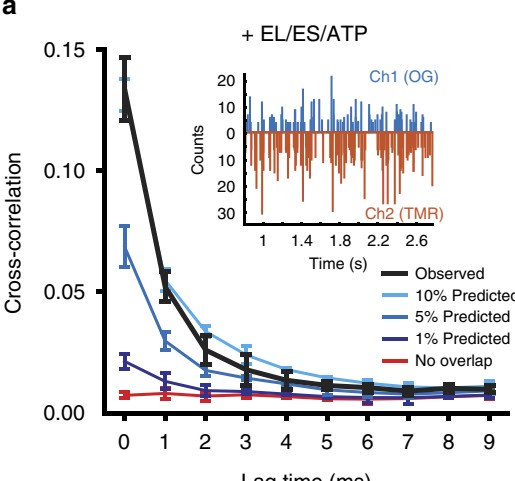

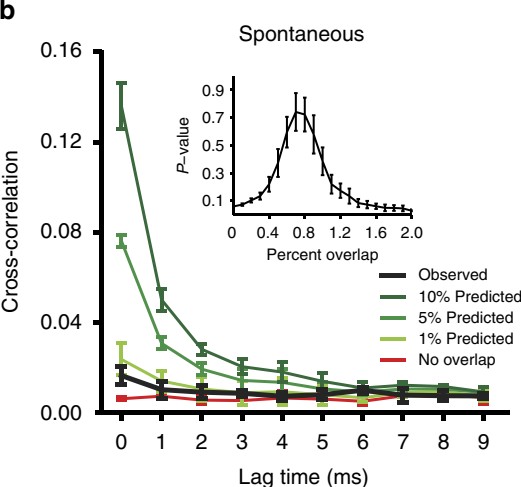

**Figure 3 | Non-native PepQ does not aggregate at very low concentrations.** (**a**) The formation of the native PepQ dimer was examined with single-molecule, two-colour co-incidence detection. Samples of PepQ-24TMR and PepQ-24OG were denatured, mixed at a stoichiometry of 1:1 (100 nM total PepQ monomer), and then refolded with GroEL, GroES and ATP for 20 min. The sample was diluted 1,000-fold and examined for fluorescence bursts. Examples of the photon history from each detection channel are shown in the inset. Fluorescence burst co-incidence was examined by cross-correlation analysis of the experimental burst data (black). The cross-correlation of numerically generated burst data with known levels of co-incidence (10%, 5%, 1% and no overlap) are also shown. (**b**) Two-colour co-incidence analysis of spontaneous PepQ folding. Samples of PepQ-24TMR and PepQ-24OG were denatured, mixed at a stoichiometry of 1:1 and directly diluted in buffer to a final monomer concentration of 2 nM. The protein was allowed to fold spontaneously at 23 °C for 10 min. The sample was then diluted 20-fold and examined for fluorescence bursts. Cross-correlation analysis of the experimental data set (black), in comparison with numerically generated burst data at known co-incidence levels, are shown. The inset illustrates the P value distribution for fitting of the experimental data to numerically generated data sets of known co-incidence, yielding a maximum co-incidence likelihood of <1%.

The stimulation of PepQ folding by the GroEL C-termini could, in principle, result from: (1) enhanced unfolding of PepQ by the C-termini[19,22,23,39]; (2) stimulation of productive folding transitions, or blockage of inhibitory ones, by the tails during intra-cavity folding[20,34,41]; or (3) a combination of both unfolding and confinement effects. Importantly, these models all predict

that the folding trajectory of a PepQ monomer inside the GroEL-GroES cavity should change upon C-terminal tail removal. To test this prediction, we exploited the tryptophan fluorescence properties of PepQ to examine a single round of encapsulation inside both the full-length SR1-GroES cavity and the truncated SRΔ526-GroES cavity. Strikingly, the rapid and early rise in tryptophan fluorescence that is observed when PepQ folds inside the SR1-GroES cavity, completely disappears

when PepQ is encapsulated inside a SRΔ526-GroES cavity (Fig. 5b). These observations are highly consistent with the idea that GroEL promotes conformations of the PepQ monomer that are not, or at least not well, populated during spontaneous folding in free solution and that the C-terminal tails are at least partially involved in this process.

**Cryo-EM observation of PepQ unfolding by the GroEL C-termini.** We previously demonstrated that GroEL helps stimulate productive folding of a kinetically trapped RuBisCO monomer through partial unfolding[19,22,23,39]. In addition, we showed that maximal RuBisCO unfolding requires the GroEL C-terminal tails[40]. If structural disruption of the misfolded substrate proteins is a general feature of GroEL-stimulated folding, then GroEL could also be expected to unfold the kinetically trapped PepQ monomer. To test this proposition, we first examined the protease susceptibility of a PepQ folding intermediate bound to both wild-type GroEL and Δ526. Chemically denatured PepQ was first bound to the open, *trans* ring of an asymmetric GroEL-GroES complex created with either wild-type GroEL or Δ526, then treated with limiting amounts of chymotrypsin[23]. Consistent with our previous RuBisCO observations, PepQ bound to a full-length GroEL ring was degraded ∼2.5-fold faster than PepQ bound to the Δ526 ring (Fig. 5c).

To develop a more detailed picture of the interaction between PepQ and GroEL, we employed cryo-EM to examine the structures of both wild type GroEL and Δ526 tetradecamers bound to non-native PepQ. Chemically denatured PepQ was first mixed with unliganded (apo) GroEL or Δ526 tetradecamers, then vitrified in thin ice and imaged with single-particle cryo-EM. Reference-free two-dimensional (2D) image classification revealed a robust population of GroEL tetradecamer complexes with substantial density visible in the central cavity of a major 2D class-average for both wild-type GroEL and Δ526 (Supplementary Fig. 4). The observed central density is highly consistent with the expected binding position of the non-native PepQ monomer. Further, three-dimensional (3D) classification and map refinement, without any applied symmetry, revealed both apo and PepQ-bound states of the tetradecamer for both wild-type GroEL and Δ526. (Fig. 6 and Supplementary Figs 5 and 6). On the basis of the gold-standard Fourier shell correlation, the overall resolution for the apo states of both wild-type GroEL and Δ526 was 7.9 Å, while the overall resolution of the pepQ-bound states for both wild-type GroEL and Δ526 was 8.3 Å (Supplementary Fig. 7A). Importantly, the resolution obtained was not uniformly distributed over the entire structure of either complex, but was

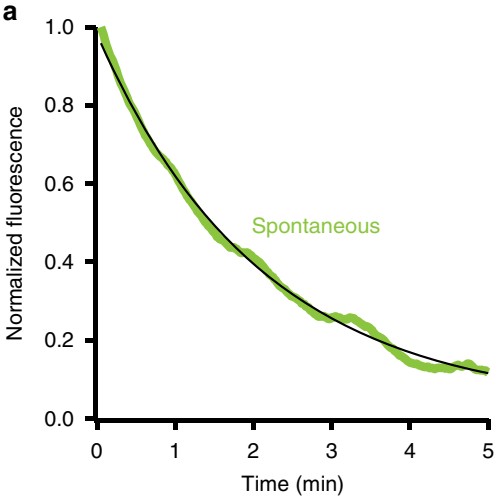

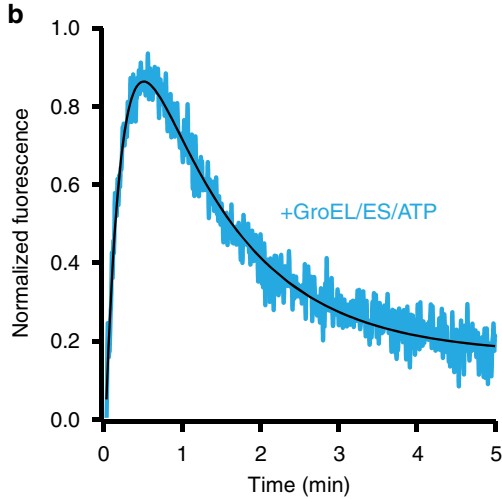

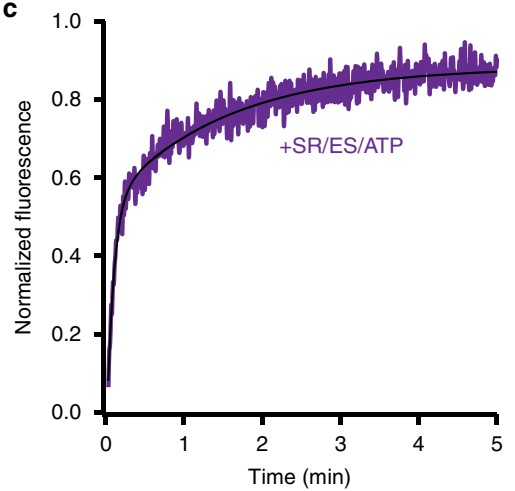

**Figure 4 | GroEL alters the folding trajectory of the PepQ monomer.**
Folding of PepQ was monitored by changes in intrinsic tryptophan fluorescence during (**a**) spontaneous folding, (**b**) folding by the fully cycling GroEL-GroES system (**c**) folding after a single round of encapsulation within the SR1-GroES complex. For spontaneous folding, wild-type PepQ was first denatured in acid-urea then diluted directly into buffer (100 nM). For GroEL-GroES folding, acid-urea denatured PepQ (100 nM) was bound to wild-type GroEL (200 nM) and refolded in the presence of GroES (400 nM) and ATP (2 mM). For SR1-GroES folding, acid-urea denatured PepQ (100 nM) was bound to SR1 (300 nM) and refolded in the presence of GroES (600 nM) and ATP (2 mM). In all cases, the traces shown represent the average of 10 independent experimental replicates. All traces were fit (solid lines) to either a single-exponential rate law (spontaneous) or a sum of exponentials (GroEL-GroES and SR1-GroES). The observed rate constants were $-0.477 \pm 0.003\,\mathrm{min}^{-1}$ for spontaneous folding, $4.63 \pm 0.05\,\mathrm{min}^{-1}$ and $-0.826 \pm 0.007\,\mathrm{min}^{-1}$ for GroEL-GroES folding and $11.5 \pm 0.2\,\mathrm{min}^{-1}$ and $0.669 \pm 0.010\,\mathrm{min}^{-1}$ for SR1-GroES folding.

significantly higher in the equatorial and intermediate domains, and lower in the apical domains (Supplementary Fig. 7B–E). The lowest local resolution was observed for density associated with the PepQ monomer in the central cavity, as expected for a non-native protein folding intermediate, which likely populates a mixture of conformations.

The cryo-EM structures reveal the striking impact of the GroEL C-termini on both the conformation and the binding position of the PepQ folding intermediate. In the absence of the C-terminal tails, the PepQ monomer appears as a strong extra density associated with the upper, inner surface of the apical domains of the GroEL cavity (Figs 6c and 8d). By contrast, on a wild-type GroEL ring, the PepQ folding intermediate shifts to a much lower average position in the cavity, moving towards the base of the cavity and in the direction of the C-termini. At the same time, the density of the PepQ intermediate decreases significantly (Figs 6f and 8c), which indicates a more unfolded and heterogenous conformational ensemble of the PepQ, leading to its weaker density in the cryo-EM map. The non-native PepQ monomer can also be seen to make contact with multiple GroEL subunits on both a wild-type GroEL and Δ526 ring (Fig. 7). However, the location of the contacts between the PepQ folding intermediate and the GroEL subunits changes dramatically when the C-terminal tails are removed. In the tailless Δ526 ring, the PepQ monomer appears to make exclusive contact with the central face of the apical domains, in the region of helices H and I (Figs 7a–c and 8d). By contrast, on a wild-type GroEL ring, the PepQ monomer shifts to a significantly lower position at the base of the apical domain, and is accompanied by a set of new, strong contacts that localize in the region of the GroEL C-terminal tails (Figs 7d–f and 8c). To confirm that the density observed at the base of the wild-type GroEL cavity does, in fact, originate from the C-termini, we examined this region in the empty wild-type GroEL and Δ526 tetradecamers. As expected, the density observed at the base of the wild-type GroEL ring, projecting from the precise position expected for the C-termini, is missing in the tailless Δ526 ring (Supplementary Fig. 8A,B). In total, these result strongly support the idea that the non-native PepQ monomer is significantly more unfolded when bound to a GroEL ring with intact C-terminal tails and this unfolding has a direct functional impact on the efficiency of productive folding.

## Discussion

Fundamentally, chaperonins like GroELS function as kinetic editors of protein folding reactions, altering how folding intermediates partition between available conformational states.

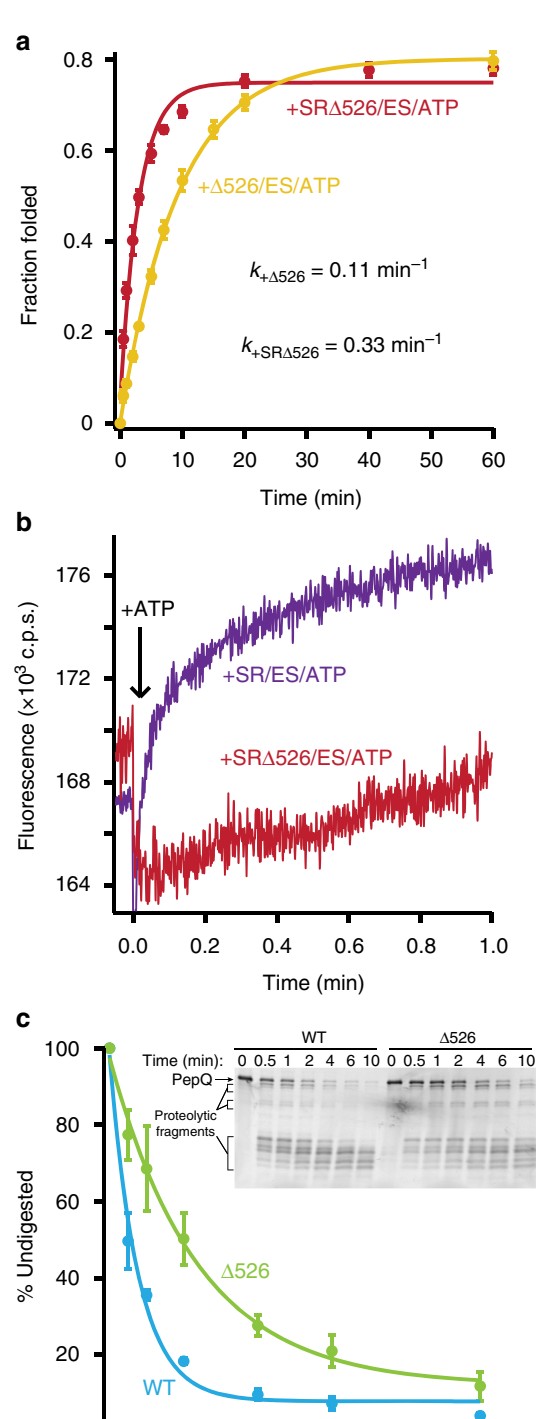

**Figure 5 | The GroEL C-termini alter the conformation and folding of the PepQ monomer.** (**a**) Acid-urea denatured PepQ was bound to a C-terminal truncation mutant of tetradecamer GroEL, Δ526 (200 nM, red) or the single-ring truncation mutant, SRΔ526 (300 nM, orange) and refolded in the presence of GroES (400 and 600 nM, respectively) and ATP (2 mM). In each case, the observed regain in enzymatic activity was fit to a single-exponential rate law (solid lines), resulting in observed rate constants of $0.106 \pm 0.003\,min^{-1}$ for Δ526-mediated folding and $0.332 \pm 0.038\,min^{-1}$ for SRΔ526-mediated folding. (**b**) Intra-cavity folding of PepQ at early times was monitored by changes in tryptophan fluorescence following addition of GroES and ATP to complexes of non-native PepQ bound to SR1 (blue) or SRΔ526 (green). Acid-urea denatured PepQ (100 nM) was first bound to either SR1 or SRΔ526 (300 nM in both cases), and then rapidly mixed with an equal volume of GroES (600 nM) and ATP (2 mM) in a stopped-flow apparatus. The traces shown represent the average of 20 experimental replicates. (**c**) Residual structure in a GroEL-bound PepQ folding intermediate was examined by protease susceptibility. PepQ-24F (100 nM) was denatured in acid-urea and bound to the *trans* ring of either wild-type GroEL-GroES or Δ526-GroES ADP complexes (ref. 23; 120 nM) and then treated with chymotrypsin for the indicated times before quenching with phenylmethylsulfonyl fluoride (PMSF) (1 mM). Samples were analysed by SDS–PAGE and laser-excited fluorescence gel scanning (inset). The migration position of full-length PepQ, as well as the position of three dominant groups of proteolytic fragments, are indicated. The amount of full-length PepQ was quantified by densitometry. The data were fit to a single-exponential rate law, with a half-time for the digestion of PepQ bound to the open ring of a wild type GroEL-GroES-ADP complex of $0.53 \pm 0.06\,min$ (EL, blue) and $1.66 \pm 0.17\,min$ for the Δ526-GroES ADP complex (Δ526, green). In all cases, error bars show the s.d. of three experimental replicates.

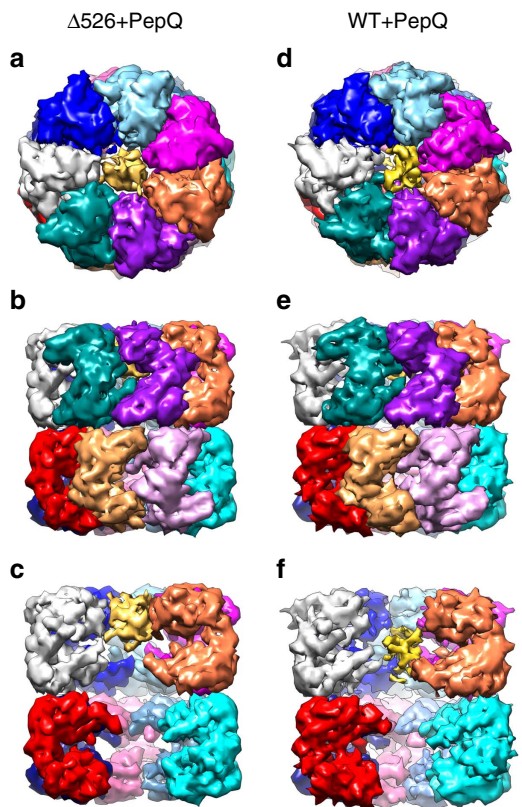

**Figure 6 | The impact of the GroEL C-termini on a bound PepQ monomer.**
The conformation of a non-native PepQ monomer bound to either a
wild-type GroEL tetradecamer (*WT*) or to the C-terminal truncation
variant (Δ526) was examined by cryo-EM. The map of the Δ526-PepQ
complex is shown from the top (**a**), side (**b**) and cutaway (**c**) views.
The map of the wild-type GroEL-PepQ complex is also shown from
the top (**d**), side (**e**) and cutaway (**f**) views. Each subunit of GroEL is
coloured differently while the PepQ monomer is coloured yellow. In the
Δ526 GroEL, the density attributed to the PepQ monomer is observed near
the top of the GroEL cavity in one ring (**c**). However, in the wild-type GroEL
complex, the PepQ density is observed near the centre of the cavity (**f**) in
one ring.

A key question, however, is whether chaperonins achieve
this editing action by actively altering the conformational
space available to their substrate proteins, or by exclusively
working as passive aggregation inhibitors. We examined this
issue from a new angle by characterizing the folding of the
*E. coli* metalloprotease PepQ, a stringent, *in vivo* GroEL-substrate
protein. We found that slow spontaneous folding of PepQ is not
caused by inhibitory aggregation. The capture of this kinetically
trapped PepQ folding intermediate by a GroEL ring results in
conformational perturbations that are consistent with unfolding.
In addition, the intrinsically unstructured C-terminal tails of the
GroEL subunits play a central role in this process (Fig. 9a).

Determining the function of the flexible C-terminal tails in
chaperonin-assisted protein folding has been challenging. Early
studies showed that the tails play no role in tetradecamer assembly
or stability[42]. At the same time, removal of the C-termini was
found to have negligible impacts on *E. coli* growth under standard
laboratory conditions, leading to the suggestion that the tails do not
play any important role in assisted folding[42,43]. Other studies,
however, demonstrated that removal of, or large alterations to, the
C-termini can have serious negative consequences in *in vitro*
protein folding assays[34,39,41,44,45]. In addition, *E. coli* strains
possessing C-terminally truncated GroEL genes display

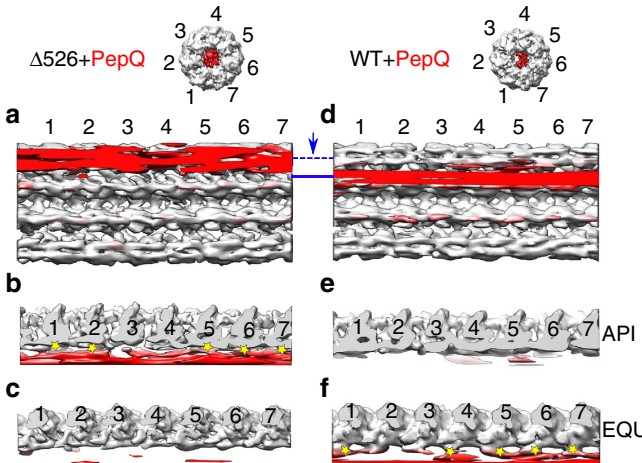

**Figure 7 | The PepQ monomer interacts with multiple GroEL subunits.**
GroEL tetradecamers with bound PepQ are shown in unwrapped, planar
displays as viewed from inside the ring for the Δ526 (**a**) and wild-type
(**d**) GroEL complexes. A top view slice of the planar map, through the
apical (API) domains (dashed blue line), is shown in **b** for Δ526 and in **e** for
wild-type GroEL. Interactions between the non-native PepQ monomer
and the Δ526 GroEL apical domains of subunits 1, 2, 5, 6 and 7 are
highlighted (**b**; yellow stars). A top view slice of the planar map, through
the equatorial (EQU) domains (solid blue line), is shown in **c** for Δ526 and
in **f** for wild-type GroEL. Contacts between the non-native PepQ and the
C-termini of subunits 1, 4, 5, 6 and 7 of the wild-type GroEL are highlighted
(**f**; yellow stars). The isosurface threshold for **b** and **c** is 1.74σ and is
1.65σ for **e** and **f**.

substantially compromised fitness in competition with wild-type
strains[42]. These observations, in combination with the extensive,
although not quite universal, conservation of the chaperonin
C-terminal tails over much of phylogeny[46,47] suggest that the
C-termini do play an important role in assisted protein folding.
Our prior work with RuBisCO supported this conclusion,
implicating the C-termini in substrate protein capture, retention
and unfolding during GroES binding[39,40]. The observations we
present here with PepQ strengthen and extend these conclusions,
showing that the unstructured C-termini make physical contact
with a non-native substrate protein before ATP or GroES binding.
In addition, we have visualized the consequences of this interaction,
demonstrating simultaneous engagement of a folding intermediate
by both the inner apical face and the unstructured tails of multiple
GroEL subunits. This multi-level binding mode both retains the
folding intermediate deeper inside the GroEL cavity and assists in
partial unfolding of the misfolded PepQ monomer.

Interestingly, we observe a single, well populated class of the
PepQ folding intermediate bound to a GroEL ring, both in the
presence and absence of the GroEL C-termini. This contrasts with
a previous cryo-EM study conducted with the the smaller
substrate protein malate dehydrogenase (MDH), in which
asymmetric model refinement suggested multiple potential
binding modes of the MDH folding intermediate[48]. While the
C-termini were not resolved in this prior study, and the
resolution of these MDH structures is several angstroms lower
than the PepQ structures we report here, two sub-populations of
the MDH folding intermediate appear to be bound in a deep
internal position within the GroEL cavity, consistent with the
binding position we observe with PepQ. A third sub-population
of the bound MDH monomer appeared to be bound in a more
elevated position near the upper, exterior surface of the GroEL
apical domains[48]. At the same time, the MDH folding
intermediate displayed a substantially smaller contact surface

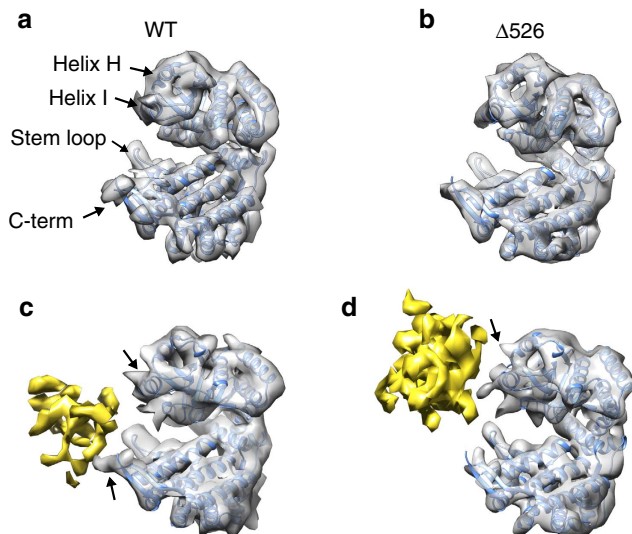

**Figure 8 | The GroEL C-termini helps retain and unfold the PepQ monomer.** (**a,b**) A single subunit of the apo GroEL atomic model (PDB ID: 4HEL) fit into the cryo-EM densities of unoccupied wild-type GroEL and Δ526 GroEL tetradecamers. The positions of the H and I helices of the GroEL apical domain, as well as the equatorial stem loop (D41-P47) and the C-terminus are labelled on the wild type GroEL structure. (**c,d**) Single subunit of the apo GroEL atomic model fit into the cryo-EM densities of PepQ-bound wild-type GroEL and Δ526 GroEL tetradecamers. The density from the non-native PepQ monomer is coloured yellow. Black arrows (**c,d**) indicate the interactions between a GroEL subunit and the PepQ monomer. When the density volumes of the Δ526 and wild-type GroEL tetradecamers are matched (∼61,000 Å$^3$ for a single GroEL subunit in both cases), the observed density volumes for the PepQ monomer are 8,564 Å$^3$ in the Δ526 and 2,696 Å$^3$ in the wild type GroEL complex, consistent with the PepQ monomer being more unfolded when bound to the wild type GroEL ring.

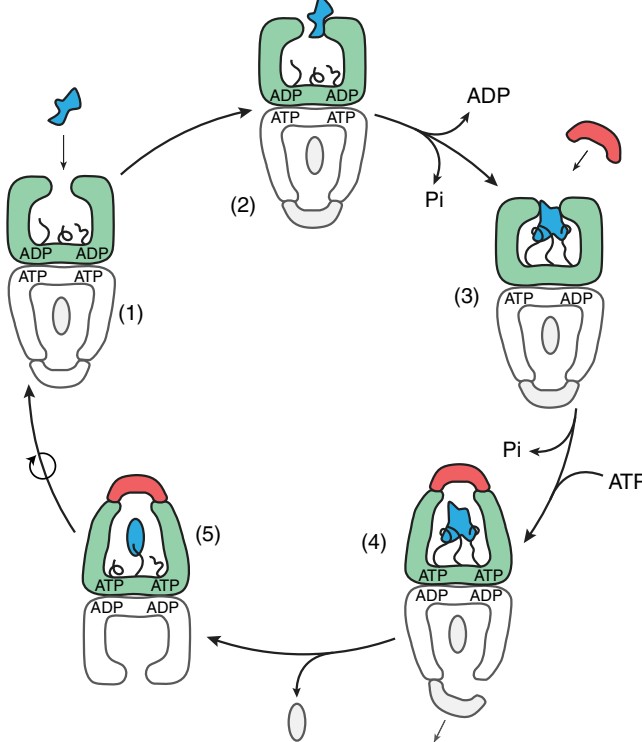

**Figure 9 | Schematic of the GroEL-GroES reaction cycle.** (1) A non-native substrate protein (irregular blue shape) enters the GroEL reaction cycle on the open *trans* ring (green) of the ATP bullet complex[22]. (2) Substrate protein binding accelerates both the release of ADP from the *trans* ring and ATP hydrolysis in the opposite, *cis* ring (grey; refs 50,51). (3) Binding of the non-native substrate protein by the C-terminal tails (black), helps retain the substrate protein deep within the GroEL cavity and, in combination with additional binding by multiple apical domains, results in substrate protein unfolding (results here and refs 19,22,24,39,40,55). (4) Assembly of the new folding cavity on the *trans* ring causes both forced unfolding and compaction of the substrate protein, and is directly coupled to the disassembly of the folding cavity on the opposite ring, potentially through a transient, symmetric intermediate[22,23,37,77-79]. (5) A subsequent allosteric shift of the GroEL-GroES complex results in full ejection of the substrate protein into the enclosed GroEL-GroES cavity and the initiation of folding before ATP hydrolysis[50]. Because ATP hydrolysis is the rate limiting step of the reaction cycle, increased binding of substrate proteins to the open *trans* ring (2) results in more rapid cycling of the GroEL-GroES system and a shorter folding cavity lifetime[22,50,51,80].

with the GroEL ring[48] in comparison to what we observe with PepQ (Fig. 7). This observation is consistent with the difference in relative mass of MDH (33 kDa) compared to PepQ (52 kDa) and suggests that the smaller MDH protein could be bound more weakly or sample a larger potential range of bound states.

In principle, the cryo-EM structure of the PepQ-GroEL complex could also reveal conformational changes of the GroEL tetradecamer that are coupled to substrate protein capture. Overall, the conformations of the GroEL tetradecamer in the presence and absence of the PepQ folding intermediate are similar. Notable breaks in the rotational symmetry of the PepQ-occupied GroEL ring, both in the presence and absence of the C-termini, are apparent (Fig. 7). However, significant deviations from ideal rotational symmetry are also observed in the unoccupied apo GroEL and Δ526 rings (Supplementary Fig. 8C–J). Strikingly, PepQ binding induces a dramatic increase in the rotational symmetry of the Δ526 apical domains, in both the bound and second, unoccupied rings (Supplementary Fig. 8H,J). The structural changes seen in the second, unoccupied ring are most likely a consequence of allosteric coupling between the GroEL rings. The coordinated binding and release of nucleotides, GroES and substrate proteins are well established and essential features of the functional GroEL reaction cycle[49]. Many of the structural details of this allosteric coupling remain poorly understood, however. In particular, it remains unclear how substrate protein binding forces ADP out of one GroEL ring while simultaneously accelerating the release of GroES from the opposite ring[37,50,51]. Previous work suggested that this allosteric

response may involve counter-clockwise movements of the GroEL apical domains, in both the substrate occupied and second, empty ring[52]. Our structural observations with PepQ suggest that a shift in the rotational symmetry of the GroEL apical domains likely also plays a role. In addition, the C-termini appear to be intimately involved in modulating this structural shift. This observation is consistent with our prior observations that removal of the C-termini attenuates negative cooperativity in ATP binding between the two GroEL rings[39].

Overall, our observations with PepQ are not consistent with an exclusively passive, aggregation-blocking role for GroEL in stimulated protein folding[53,54]. These observations are, however, fully consistent with our previous demonstration that GroEL plays an active role in the assisted folding of *R. rubrum* RuBisCO (refs 19,22,23,39). They are also consistent with observations from other groups on other substrate proteins, including another endogenous *E. coli* enzyme DapA[20,21,24,25,55,56]. Our observations

with PepQ also suggest that active participation by GroELS in stimulated protein folding is likely to be a general mechanistic feature of these chaperonin machines. DapA, like RuBisCO, is a member of the TIM-barrel family of proteins, a canonical α/β-fold that is highly represented in the subset of *E. coli* proteins that depend on GroEL for folding[31,32]. By contrast, PepQ is a member the so-called pita-bread proteins[28–30], a protein fold that is fundamentally distinct from the TIM-barrel fold[27]. To date, no pita-bread fold has been examined in detail as a GroEL-substrate protein. The addition of PepQ to the list of *E. coli* proteins that derive a large, active folding enhancement from GroEL strengthens the argument that similar mechanisms are likely to stimulate the folding of many stringent substrate proteins.

Interestingly, PepQ appears to have no ready access to fast and productive folding pathways in free solution. At the same time, persistent misfolding produces PepQ monomers that, although they do not aggregate, cannot reach the native state even with assistance from GroEL. This suggests that the conformational search of the non-native PepQ monomer, at least in free solution, is dominated by deep and inhibitory kinetic wells that GroEL helps the protein to avoid. Whether the iterative annealing or confinement-based models most accurately describe this active folding mechanism of GroEL remains controversial[5]. Importantly, these mechanisms make distinct predictions about what should happen to PepQ folding when the GroEL cycling rate is altered. If unfolding of kinetically trapped intermediates is important for stimulated folding of PepQ, it should be possible for the cycling GroEL-GroES system to achieve a stimulated folding rate that exceeds the limiting, intra-cavity folding rate observed with SR1-GroES. By contrast, if confinement is most important for PepQ folding, then the non-cycling SR1-GroES system should display the maximum possible enhanced folding rate, a rate that the cycling system could approach but never exceed[22].

To test these predictions, we examined the folding rate of PepQ under conditions where the GroEL-GroES cycling rate was systematically increased. Modulation of the GroEL ATPase rate was accomplished by addition of bovine serum albumin (BSA), which interacts only weakly with GroEL. Because progression of the GroEL ATPase cycle is linked to ADP release, which is in turn coupled to binding of proteins to the open, post-hydrolysis *trans* GroEL ring (Fig. 9 and refs 37,50,51), BSA can be used to accelerate the GroEL-GroES ATPase cycle (Fig. 10a). However, because the interaction between BSA and GroEL is weak, BSA only poorly competes with PepQ for binding to GroEL. At concentrations up to 0.1 mg ml$^{-1}$, BSA has a small, negative impact on the observed PepQ folding rate observed with cycling GroEL-GroES (Fig. 10b). Strikingly, however, while the addition of BSA has no impact on either spontaneous PepQ folding or SR1-GroES mediated folding, higher BSA concentrations substantially enhance the PepQ folding rate achieved with cycling GroEL-GroES (Fig. 10b). Importantly, the magnitude of this effect increases as the concentration of BSA increases, mirroring the impact of BSA on the steady-state rate of ATP turnover by GroEL-GroES (Fig. 10b). This response is very similar to our prior observations with RuBisCO (ref. 22), where an ~40% increase in the steady-state GroEL-GroES ATPase rate yielded a 2.5–3-fold enhancement of the observed RuBisCO folding rate. These observations suggest that repetitive unfolding by GroEL is required to achieve maximally stimulated folding of many stringent substrate proteins.

When considering the stimulatory impact of partial unfolding, it is important to note that GroEL unfolds substrate protein in two distinct phases. The first is associated with the capture of a folding intermediates by the GroEL ring, where a binding-driven expansion of the substrate protein can, in some cases, result is substantial conformational disruption (this study and refs 19,22–24,39,55). This unfolding event occurs both during and

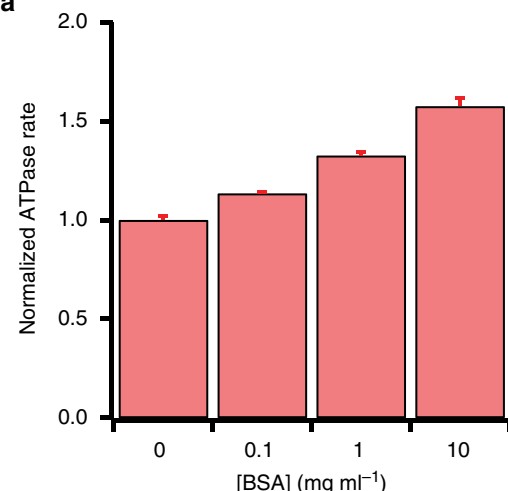

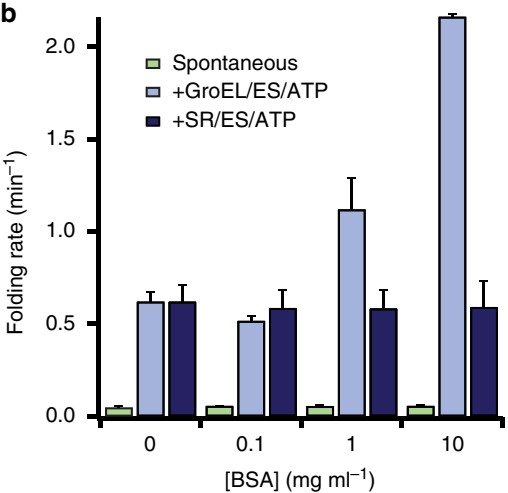

**Figure 10 | Cycling GroEL-GroES can fold PepQ faster than confinement alone.** (**a**) The rate of ATP hydrolysis by GroEL-GroES is stimulated in the presence of BSA. The steady-state rate of ATP hydrolysis by GroEL (200 nM) in the presence of GroES (400 nM) and ATP (2 mM) was measured with varying concentrations of [BSA]. Error bars show the standard deviation of three independent experiments. (**b**) Addition of BSA to a cycling GroEL-GroES system substantially accelerates the rate of assisted PepQ folding. The rate of spontaneous PepQ folding (green), intra-cavity folding with SR1-GroES (dark blue), and folding with fully the cycling wild type GroEL-GroES system (light blue) was examined in the presence of different concentrations of BSA. Experimental conditions were the same as Fig. 1, with the exception that native BSA was present in the buffer. Error bars show the s.d. of three experimental replicates.

immediately after capture of a folding intermediate, but before ATP binding. Most likely, binding-associated unfolding is similar to surface-catalysed denaturation, where the substrate protein becomes splayed across the multiple interaction surfaces of the apical domains as well as, we suggest, the C-termini[19,22,23,39,55].

GroEL also imposes a second, directed unfolding process that is impelled by ATP[22–24,39]. When a GroEL ring binds ATP, the apical domains are driven through a large-scale, rigid body rearrangement that both rotates and elevates them[57–59]. While these shifts are necessary for GroES binding and substrate encapsulation[9–12], previous observations have also demonstrated that (1) substrate proteins remain associated with the apical

domains as they initiate their movement, imposing a substantial load on their motion[60] and (2) apical domain movement can simultaneously impart a rapid, forced unfolding event on the substrate protein[22–24,39]. While our studies with PepQ were not designed to detect forced unfolding, it is striking that both binding-driven unfolding of PepQ and RuBisCO (this study and ref. 39) and forced unfolding of RuBisCO[39] are attenuated when the C-termini are removed. These observations suggest that the C-termini represent a secondary binding platform at the base of the GroEL cavity that is important both for the initial capture and unfolding of the substrate protein, as well as retention of the folding intermediate within the GroEL cavity during the process of apical domain movement and GroES binding[40]. It remains an open question how the C-termini are induced to release the substrate protein upon the initiation of folding. However, both experimental[39] and computational studies[61] indicate that the C-termini are coupled to the GroELS allosteric cycle, suggesting that modulation of the interaction between the C-termini and a folding intermediate might be controlled by the GroEL ATPase cycle in a manner that parallels the behaviour of the apical domains.

Fundamentally, the iterative annealing and confinement mechanisms are not mutually exclusive. A combined mechanism, in which kinetically trapped folding intermediates are first partially unfolded, then briefly confined within the privileged environment of the GroEL-GroES cavity where re-population of misfolded conformations is discouraged, might well yield a maximally efficient strategy for accelerating the folding of especially recalcitrant proteins. Several of our observations with PepQ are consistent with such a mechanism. In the presence of either single-ring or double-ring GroEL variants, PepQ displays a sizable fluorescent burst phase that is completely absent during spontaneous folding (Fig. 4). This observation suggests that the PepQ monomer, while confined within the GroEL-GroES cavity, populates at least one conformational state (or ensemble of states) with ready access to the native state. During spontaneous folding; however, this state is either very rarely populated, or not populated at all. At the same time, removal of the GroEL C-terminal tails slows overall PepQ folding and completely eliminates the fluorescence burst phase (Fig. 5). This behaviour is strikingly similar to the impact of C-terminal tail removal on RuBisCO folding, where the formation of a rapidly folding, burst phase intermediate depends upon both partial unfolding and encapsulation within the GroEL-GroES cavity[23,39,40]. As with RuBisCO, C-terminal tail removal also has a more profound impact on PepQ folding with the cycling, tailless Δ526 tetradecamer than it does on the tailless single-ring SRΔ526 (Figs 1 and 5). For both RuBisCO and PepQ, however, long-term confinement within the chaperonin cavity, even when partial unfolding is reduced through C-terminal tail removal (for example, SR1-GroES versus SRΔ526-GroES) results in substantially enhanced folding in comparison to the free solution folding of both proteins (Figs 1 and 5 and refs 3,20,39). In total, these observations are consistent with an active chaperonin mechanism in which partial unfolding and confinement lead to optimal stimulation of folding for highly dependent substrate proteins. It is worth noting that in a living E. coli cell, additional chaperone systems (for example, the Hsp70s and Hsp100s) can engage a folding intermediate before its processing by GroELS[1]. Learning how these additional chaperone systems impact the folding of GroELS substrates will be an important next step towards understanding the mechanism of chaperone and chaperonin-mediated folding pathways.

## Methods

**Bacterial strains.** All bacterial strains used in this work were originally obtained from the laboratory of Dr. Arthur Horwich at Yale University Medical School.

Bacterial strains employed:
BL21—*E. coli* B dcm ompT hsdS(rB-mB-) gal
BL21DE3—*E. coli* B dcm ompT hsdS(rB-mB-) gal [λDE3]
DH5α—*E. coli* fhuA2 lac(del)U169 phoA glnV44 Φ80' lacZ(del)M15 gyrA96 recA1 relA1 endA1 thi-1 hsdR17.

**Proteins.** Wild type and variants of GroEL (SR1 and C-terminal truncation mutants), GroES and wild type *E. coli* PepQ were all expressed and purified as described previously[19,22,23,39,50]. The cysteine mutant of PepQ, A24C, was generated via standard site-directed mutagenesis[62] and the sequence was verified by DNA sequencing. This mutant was expressed and purified following the protocol for wild type PepQ.

In brief, GroEL was expressed from an inducible plasmid in *E. coli* BL21 in LB at 37 °C. After cell disruption, the crude lysate was clarified by ultracentrifugation (142,000 × g), followed by anion exchange chromatography (FastFlow Q, GE) at pH 7.4. GroEL fractions were concentrated by 70% (w/v) ammonium sulfate precipitation. This precipitate was solubilized and dialyzed against buffer at pH 6.8 containing 25% (wild-type GroEL) or 12.5% (all GroEL mutants) methanol. A second round of strong anion exchange (FastFlow Q, GE), run in the same methanol-containing buffer at pH 6.8, was used to strip co-purifying small proteins and peptides from the GroEL oligomers. To further remove contaminating proteins and peptides that remain tightly associated through prior stages of purification, GroEL fractions were gently agitated in the same methanol-containing buffer and Affi Blue Gel resin overnight at 4 °C under an argon atmosphere. The final sample was dialysed into storage buffer (pH 7.4), supplemented with glycerol (15–20% v/v), concentrated, and snap frozen using liquid N₂.

GroES was expressed from an inducible plasmid in *E. coli* BL21(DE3) in LB at 37 °C. After cell disruption, the crude lysate was clarified by ultracentrifugation (142,000 × g), followed by acidification with sodium acetate, and anion exchange chromatography at pH 4.6 (FastFlow Q, GE). The sample was dialysed against buffer at pH 7.4 and applied to a strong anion exchange column (Source Q, GE). GroES was eluted with NaCl and enriched fractions were pooled. The sample was dialysed into storage buffer (pH 7.4), supplemented with glycerol (15–20% v/v), concentrated and snap frozen using liquid N₂.

PepQ and PepQ mutants were expressed from an inducible plasmid in *E. coli* BL21(DE3) in LB at 37 °C. After cell disruption, the crude lysate was clarified by ultracentrifugation. The supernatant was applied to a strong anion exchange column (FastFlow Q, GE) at pH 7.4 and eluted with a gradient of NaCl. Fractions enriched for PepQ were pooled, and the protein was precipitated with 70% (w/v) ammonium sulfate. The sample was loaded on a hydrophobic interaction column (Phenyl Sepharose FF, GE) at pH 7.4 and eluted with a decreasing ammonium sulfate gradient. Fractions enriched for PepQ were pooled, dialysed into storage buffer (pH 7.4), supplemented with glycerol (15–20% v/v), concentrated and snap frozen using liquid N₂.

**Labelling of PepQ.** A24C PepQ was labelled using either 5-iodoacetamido-fluorescein (fluorescein, F), 5-(2-acetamidoethyl) aminonaphthalene 1-sulfonate (EDANS, ED), tetramethylrhodamine-5-iodoacetamide dihydroiodide (tetramethylrhoadmine, TMR), or Oregon Green 488 iodoacetamide (Oregon Green, OG). All dyes were obtained from Invitrogen (Molecular Probes). PepQ (∼10 mg ml⁻¹ in 50 mM Tris, pH 8, 100 mM KCl, 1 mM MgCl₂) was reduced with 0.5 mM tris(2-carboxyethyl)phosphine (TCEP) TCEP and labelled with a 10-fold excess of reactive dye, added in 1 addition for 3 h at 23 °C. The reaction was quenched by adding glutathione (5 mM), and the labelled PepQ was first separated from unbound dye by gel filtration (PD-10 column, Pharmacia), followed by re-purification of the labelled protein with high-resolution ion exchange chromatography (MonoQ, GE). The extent of labelling was determined by protein quantification by the Bradford assay (Bio-Rad) and dye quantification under denaturing conditions using known dye molar extinction coefficients[37,63]. Unique labelling of a single cysteine was verified by both denaturing anion exchange chromatography (MonoQ, GE) in 8 M urea buffer and by detection of a single major and fluorescent tryptic peptide upon separation by C8 reverse-phase chromatography[63].

**Folding assays.** PepQ refolding assays were conducted in TKM buffer (50 mM Tris-HCl, pH 7.4, 50 mM KOAc, 10 mM Mg(OAc)₂ and 2 mM DTT) using a protocol similar to that employed previously for RuBisCO (refs 19,22,23,37,50,63), with differences in the folding buffer composition, duration of post-reaction incubation, and the detailed assay method[27]. All folding assays were conducted using PepQ that was diluted at least 40-fold into 8 M urea, 25 mM glycine phosphate, pH 2, and incubated at room temperature for at least 20 min before further use. CD spectra show a complete loss of secondary structure under these conditions (Supplementary Fig. 9). Spontaneous refolding of PepQ was initiated by a 50-fold dilution from denaturant into TKM buffer (50 mM Tris-HCl, pH 7.4, 50 mM KOAc, 10 mM Mg(OAc)₂ and 2 mM DTT) and quenched through the addition of excess GroEL. Chaperonin-mediated folding reactions using either wild-type or mutant tetradecameric GroEL began with a 50-fold dilution of denatured PepQ into TKM buffer containing a particular GroEL variant. GroES and ATP were added to initiate the reaction cycle and the reaction was quenched

with hexokinase and glucose[22,23,39]. Folding reactions in single-ring mutants of GroEL were done similarly, except quenching was accomplished by the simultaneous addition of EDTA and incubation of the sample at 0 °C (refs 20,34). After quenching, all samples were incubated for 60 min at room temperature to allow for dimerization. The enzyme activity of all samples was measured through an NAD-coupled reaction using alanine dehydrogenase from *B. subtilis*[27].

**Measuring PepQ persistence in solution.** Fluorescein-labelled PepQ (24F) was allowed to refold spontaneously or in the presence of the chaperone system (as in Fig. 1b, see Folding assays in Methods section). Samples were taken after 60 min and run on 10% SDS–polyacrylamide gel electrophoresis (SDS–PAGE). Gels were imaged with a Typhoon Trio (GE Healthcare) and quantified with ImageJ.

**Fluorescence and light scattering.** Light scattering and fluorescence measurements were conducted with a T-format fluorometer (PTI), equipped with a jacketed cuvette holder for temperature control with a high-precision circulating water bath (Neslab). For both types of experiments, the assays were initiated by diluting acid-urea denatured PepQ at least 50-fold into temperature-equilibrated TKM buffer (23 °C) in the presence or absence of GroEL. Tryptophan fluorescence was monitored with excitation at 295 ± 4 nm and emission at 340 ± 4 nm. The excitation and emission wavelengths were both 340 ± 1 nm for light scattering experiments.

**Stopped-flow fluorescence.** Stopped-flow experiments were conducted using an SFM-400 rapid mixing unit (BioLogic) equipped with a custom-designed two-channel fluorescence detection system[19,23,39,63]. Mixing was done using equal volume injections from two syringes, one containing GroEL-PepQ complexes and one containing GroES and ATP. Each solution was in TKM buffer. Measurements were taken every 150 ms.

**Steady-state FRET.** Steady-state fluorescence measurements were conducted with a with a T-format fluorometer (PTI), equipped with a jacketed cuvette holder for temperature control with a high-precision circulating water bath (Neslab). FRET efficiencies were calculated from the changes in donor-side fluorescence of matched donor only and donor plus acceptor labelled molecules[37,63].

**Single-molecule fluorescence microscope.** Built on a research quality, vibrationally isolated 4′ × 8′ optical table, the system is constructed around a Nikon Eclipse Ti-U inverted microscope base using a × 60/1.4NA CFI Plan Fluor oil immersion objective. The microscope base is outfitted with a precision, 2-axis stepper motor sample stage (Optiscan II; Prior) and a custom-designed confocal optical bench with three, independent detection channels. Each detection channel is configured with an optimized band-pass filter set for wavelength selection and a low-noise, single photon counting APD unit (SPCM-AQRH-15; Excelitas). Photon pulses are collected and time stamped with either a multichannel hardware correlator (correlator.com) or high speed TTL counting board (NI9402; National Instruments). Sample excitation is provided by either one or a combination of three lasers: two diode lasers (488 and 642 nm; Omicron) and one diode-pumped solid state laser (561 nm; Lasos). The free-space beams of each laser are each coupled to a three-channel fibre combiner (PSK-000843; Gould Technologies) and the combined output is directed into the sample objective with a custom, triple-window dichroic filter (Chroma). Each laser is addressable from the integrated control and data acquisition software, custom developed using LabView (National Instruments).

**PepQ refolding by fluorescence correlation spectroscopy.** PepQ-24TMR was diluted greater than 40-fold (to 5 μM) into 8 M urea, 25 mM glycine phosphate, pH 2 and incubated for 20 min at room temperature. For spontaneous folding reactions, this PepQ-TMR was then diluted to 100 nM in the same solution. The folding reaction was initiated by dilution of PepQ to 2 nM in TKM buffer. Folding was quenched by the addition of 50 μl of the refolding reaction to 50 μl of 1 μM GroEL in TKM buffer. For GroEL-mediated folding, 5 μM denatured PepQ-24TMR was diluted to 100 nM in TKM buffer containing GroEL (200 nM final tetradecamer concentration). After a 10 min incubation at room temperature, this solution was diluted into TKM buffer containing GroEL, GroES and an ATP-regeneration system[22]. Folding was then initiated by the addition of ATP. The final concentration of ATP was 2 mM, GroEL was 1 μM, and GroES was 2 μM. Folding was quenched by the addition of 20 μl of the reaction mixture with an equal volume of hexokinase and glucose. Dimerization was not observable in refolding assays conducted at 1–2 nM PepQ, based on a reproducible lack of detectable enzymatic activity, even with up to eight hours of incubation at 23 °C. PepQ enzymatic activity is, however, detectable when the native dimer is diluted to 1–2 nM (Supplementary Fig. 10). Fluorescence correlation spectroscopy (FCS) data were collected by placing 10 μl of the quenched reaction mixtures onto BSA-blocked coverslips mounted on a custom-built, inverted confocal microscope and covered with a humidified chamber to prevent evaporation.

Autocorrelation curves were collected for 2 min using 50 μW continuous input power from a 561 nm diode-pumped solid state laser. Autocorrelation curves were normalized in mean amplitude between $10^{-6}$ and $10^{-5}$ s for display purposes. As standards, the autocorrelation curves of PepQ fully bound to GroEL (obtained by not adding ATP to a folding reaction), as a native dimer (obtained by diluting native PepQ-24TMR in buffer), and as a native monomer (obtained by allowing a GroEL-mediated folding reaction with 1 nM PepQ to continue for an hour) were also determined. Each autocorrelation curve was fit using a multi-component model[64,65] to account for populations of freely diffusing and GroEL-bound PepQ. Curve fitting was conducted using two different approaches. First, the diffusion coefficient of each population was fixed and the fractional population was allowed to vary. Second, the average diffusion coefficient of the entire population was determined. The resulting refolding curves obtained from these two methods were statistically equivalent.

**Two-colour single-molecule co-incidence detection.** The 24OG and 24TMR PepQ variants were each diluted to 5 μM in acid-urea and allowed to unfold at room temperature. For spontaneous folding, the two solutions were diluted together to a concentration of 50 nM each. Folding was initiated by a 50-fold dilution into TKM buffer containing 0.1 mg ml⁻¹ BSA. BSA has no effect on the folding rate of PepQ (Fig. 10b), but was necessary to prevent loss of protein at very-low concentrations to liquid handling equipment. After 10 min at 23 °C, samples were diluted 20-fold into the same buffer and 10 μl samples were immediately placed on a BSA-blocked, optical glass coverslip mounted in a custom holder, fitted on the microscope stage. Samples were covered with a humidifier cap to prevent evaporation. For chaperone-mediated folding, the two 5 μM solutions of denatured, labelled PepQ were diluted together to 50 nM each into TKM containing GroEL and GroES, and folding was initiated by the addition of ATP (2 mM) to a solution containing: 50 nM PepQ-24OG, 50 nM PepQ-24TMR, 200 nM GroEL, 400 nM GroES). After incubated at 23 °C for at least one hour, the reaction was halted by a 50-fold dilution into TKM. This sample was then diluted a further 20-fold into TKM containing 0.1 mg ml⁻¹ BSA and immediately assayed. Fluorescence burst data were collected for each sample over a 1 min window using 100 μs sampling bins. Simultaneous excitation was provided from two co-aligned lasers (488 and 561 nm), each providing 200 μW of power at the sample.

To quantify the formation of native PepQ dimers resulting from productive GroEL folding, as well as the formation of PepQ aggregates during spontaneous folding, we developed a cross-correlation statistic (plotted in Fig. 3) that evaluates the percent photon arrival time overlap between two time streams. To begin, each time stream was normalized so the maximum spike intensity amplitude had a value of one. A threshold filter was then applied (5*r.m.s.) to both colour channels to isolate spike activity and remove low-level detector noise. The filtered time streams were used to create a binary mask of spike events. On the basis of a particle transit time through the excitation volume of about 1 ms, both binary time streams were re-binned in 1 ms bins. The cross-correlation versus time lag between two time streams $T_1$ and $T_2$, each with a total of $N_1$ and $N_2$ non-zero time bins, was then generated for the + EL/ES/ATP and spontaneous activity data:

$$CC(lag) = \sum \frac{T_1(t-lag)*T_2(t)}{\sqrt{N_1}\sqrt{N_2}} \qquad (1)$$

With this normalization, the autocorrelation of any time stream had a value of 1, while the minimum cross-correlation value was bounded at zero. Due to the non-zero probability of photons randomly arriving at two detectors at the same time, the minimal cross-correlation value was not zero.

To assess our cross-correlation measure we used the photon arrival data from either + EL/ES/ATP or spontaneous PepQ activity to generate an expected baseline activity (that is, zero significant co-incidence). Each time stream was compared to a 5 s cyclically shifted version of itself to examine the correlation between two nominally uncorrelated time streams of identical photon rate and noise (denoted as No Overlap, Fig. 3c,d). In both of these baseline cases, there is approximately a 1% cross-correlation independent of lag time or detector channel (red line). We then calibrated the cross-correlation statistic for several data streams of a known and fixed amount of similarity. To do this, we replaced a segment of a time stream T1 with an equal length segment of a time stream T2 at a random location. The ratio of the segment length to total length then corresponded to the percent overlap. The original data stream T2 and the altered data stream T1 then represented two data streams of known overlap and whose cross-correlation could be used for comparison. This process was repeated 20 times for each percent overlap. The cross-correlation results were averaged and the uncertainty in the mean for each lag was monitored. The resulting family of cross-correlation curves (Fig. 3a,b) was then used to assess the level of overlap between two-colour channels that had not been shifted in time.

We then tested the null hypothesis that the pairwise differences between the cross-correlations values of the spontaneous data and various possible overlap simulations had a mean equal to zero. The resulting $P$ values from this family of t-tests indicated the most likely zero null hypothesis occurred for an overlap of 0.75% (Fig. 3b, inset).

**Triplet state conversion of fluorescent PepQ variants.** Native, TMR- or Oregon Green-labelled PepQ dimer (24TMR and 24OG, respectively) was diluted to 1 nM (dimer) in TKM buffer with supplemental BSA (0.1 mg ml⁻¹). An amount of 10 μl

of sample was pipetted onto a BSA-blocked coverslip, mounted as described for Fig. 2, using a 561 nm laser for 24TMR and a 488 nm laser for 24OG. For each sample, data were collected for 2 min, with 500 μs collection bins, at each laser power level. The entire, non-normalized data set for each dye (Supplementary Fig. 2) was fit globally using IgorPro (Wavemetrics) to an autocorrelation function that included a correction factor for the effect of a triplet state population[66,67].

**Protease protection.** The protease sensitivity of non-native PepQ bound to a GroEL ring was conducted as described previously for the substrate protein RuBisCO[23,50]. Briefly, denatured PepQ (100 nM) labelled at position 24 with fluorescein (PepQ-24F) was bound to asymmetric GroEL-GroES-ADP bullets (120 nM)[23]. Chymotrypsin (0.3 mg ml$^{-1}$) was added, and time points were taken, with the reaction stopped by addition of phenylmethylsulfonyl fluoride (PMSF) (1 mM). Samples were run on 10% SDS–PAGE and imaged using a Typhoon Trio (GE Healthcare).

**GroEL ATPase activity.** The ATPase activity of GroEL was assayed using an NADH-coupled reaction. In brief, the GroEL (200 nM) ATPase cycle was monitored in the presence of ATP (2 mM) and GroES (400 nM) using 10 U ml$^{-1}$ pyruvate kinase, 10 U ml$^{-1}$ lactate dehydrogenase, 1 mM phosphoenolpyruvate and 0.2 mM NADH. This system regenerates ATP, maintaining it at a constant concentration, and produces a decrease in the absorbance at 340 nm as NADH is consumed[22,39,50,68,69]. The rate of spontaneous ATP hydrolysis under each condition without GroEL was also determined to control for effects on the coupling system.

**Cryo-electron microscopy.** PepQ was denatured in 8 M urea, 25 mM glycine phosphate, at pH 2 and incubated for 30 min at room temperature. Denatured PepQ (50 μM), in droplets of 4.6 μl (2.3 μM per addition) was titrated into solutions of either GroEL or Δ526 (8 μM tetradecamers, 100 μl) in TKM buffer, followed by rapid, repeated mixing and then incubated at room temperature for 5 min. The final concentration of PepQ was 7 μM. 3 μl of this PepQ/GroEL mixture was applied to a C-Flat 1.2/1.3 400 mesh holey carbon grid at 20 °C with 100% relative humidity and vitrified using a Vitrobot (Mark III, FEI company, Netherlands). The thin-ice areas that showed clear and mono-dispersed particles were imaged under an FEI Tecnai F20 electron microscope with a field emission gun (FEI company, Netherlands) operated at 200 KV. Data were collected on a Gatan K2 Summit direct detection camera (Gatan, Pleasanton CA) in electron counting mode[70] at a nominal magnification of ×19,000, yielding a pixel size of 1.85 Å. The dose rate was 10 e$^-$ pixel$^{-2}$ s$^{-1}$ at the camera. A 33-frame movie stack was recorded for each micrograph, for a total exposure time of 6.6 s. The total dose onto the specimen was 19 e$^-$ Å$^{-2}$.

**Image processing and map visualization.** For the wild-type GroEL-PepQ complex, 1,450 micrographs were collected and aligned iteratively and filtered based on electron dose using Unblur[71]. Sum images (1,109) showing strong power spectra were selected for particle picking in EMAN2 (ref. 72), yielding 217,317 particles with a box size of 160 × 160 pixels[2]. These particles were processed in Relion[73] with a downscaling factor of 2, respectively. Two runs of reference-free 2D classification were performed to produce a 'cleaner' data set containing 170,639 particles, which were separated into four classes in 3D classification with no symmetry applied. Particles belonging to apo GroEL and PepQ-bound GroEL were used for final asymmetric 3D refinement, respectively (Supplementary Fig. 5). The processing procedure used for the Δ526-PepQ complex was the same as used for the wild type GroEL complex. Briefly, 847 movie stacks were collected and 703 aligned images were picked, yielding 224,696 particles. 117,040 clean particles were screened after two runs of reference-free 2D classification. 3D classification and asymmetric refinement was performed similar as wild-type GroEL-PepQ complex (Supplementary Fig. 6). The final resolutions of the 3D density maps for both wild-type GroEL and Δ526 were 7.9 Å for the apo states and 8.3 Å for the PepQ-bound states (Supplementary Fig. 7A), assessed with the gold-standard criterion at 0.143 Fourier shell correlation[74]. Local resolutions were estimated using Blocres[75]. The unwrapping of the maps was done with 'e2unwrap3d.py' in EMAN2. Visualization and fitting of atomic models into the cryo-EM density maps, were done in UCSF Chimera[76].

**Circular dichroism spectroscopy.** PepQ was diluted >100-fold into 25 mM sodium phosphate, pH 7.2, 100 μM MnCl$_2$ (native) or 25 mM sodium phosphate, pH 2.1, 8 M Urea (denatured) to a concentration of 0.1 mg ml$^{-1}$. Following an equilibration at 23 °C for 15 min, samples were loaded into a 1 cm path length cuvette and the circular dichroism (CD) of the sample measured in the far ultra-violet region using an Aviv 202:CD spectrometer (Aviv Biomedical). The sample temperature was equilibrated in the spectrometer to 25 °C before the initiation of measurements and was maintained at this temperature throughout. The CD signal at each wavelength was averaged for 30 s, using 1 nm wavelength steps. Scans of each sample buffer were used as blanks.

**Encapsulation of PepQ by GroEL.** PepQ-24F (100 nM) was denatured in 8 M urea, 25 mM glycine phosphate, pH 2 and bound to wild-type or Δ526

GroEL-GroES-ADP bullets (200 nM) or full-length or Δ526 single-ring GroEL (300 nM) supplemented with GroES (600 nM). A single turnover was initiated by the addition of ATP (2 mM) followed by quenching with hexokinase and glucose after 10 s. Unencapsulated PepQ was digested with Proteinase K (0.5 μg ml$^{-1}$) for 10 min before quenching with PMSF (1 mM). Samples were run on SDS–PAGE and scanned for fluorescein fluorescence using a Typhoon Trio (GE Healthcare). Band intensity was measured with ImageJ.

**Data availability.** The cryo-EM density maps are deposited to EMDataBank (http://www.emdatabank.org/) with accession id EMD-8316 (wild type GroEL with PepQ bound), EMD-8317 (wild type GroEL), EMD-8318(C-terminal truncated GroEL, Δ526, with PepQ bound) and EMD-8319(C-terminal truncated GroEL, Δ526). All other data are available from the corresponding authors upon reasonable request.

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

## Acknowledgements

This research has been supported by the National Institutes of Health grant (H.R. GM065421). We thank the Microscopy and Imaging Center at Texas A&M University for providing instrumentation for cryo-EM data collection. We acknowledge the Texas A&M High Performance Research Computing Center for providing the computational resources for the data processing. We also thank Dr Chavela Carr for comments and editorial assistance in preparing the manuscript. J.Z. is supported by startup funding from the Department of Biochemistry and Biophysics at Texas A&M

University and Center for Phage Technology jointly sponsored by Texas AgriLife and Texas A&M University. J.Z. is also supported by Welch Foundation grant A-1863.

## Author contributions

Conceptualization: H.S.R., J.Z. and J.W. Methodology: H.S.R., J.Z., J.W., A.R. and J.P. Formal analysis: J.P., J.Z and M.J. Investigation: J.W., A.R. and M.J. Writing—original draft: H.S.R. Writing—review and editing: H.S.R., J.Z., J.P., J.W., A.R. and M.J. Visualization: H.S.R., J.Z., J.W., J.P. and M.J. Supervision: H.S.R and J.Z. Project management: H.S.R. Funding acquisition: H.S.R and J.Z.

## Additional information

**Competing interests:** The authors declare no competing financial interests.

