## [Peer review file · Nature Communications]

Reviewers' comments:

Reviewer #1 (Remarks to the Author):

Remarks for the authors

The manuscript of Weaver et al addresses an important question with a large number of biophysical techniques with admirable rigor. For the most part I think their conclusions are justified by the experimental data (some exceptions to that below) so I conclude that it merits publication, subject to some revisions.

Lines 227-230: The argument outlined here is weak. If monomers don't form dimers at 2nM, why would they form polymers (aggregates)?

Figures 1 and 3: There is a discrepancy between the data in Fig 1B and the trp fluorescence data in Fig 3A and 3B. The t_{1/2} of spontaneous folding of PepQ is given as ~ 15 min in Fig 1B, But the t_{1/2} of the spontaneous trp fluorescence change is about 1 min.

Conversely the t_{1/2} for the GroEL/ES/ATP folding is given as ~1 min in Fig 1B, but the t_{1/2} for the decrease in trp fluorescence is ~2min. Can the authors comment on this?

Figure 3B and 3C: The data has been normalized. Are the real amplitudes the same for 3B (a 14 mer) and 3C (a 7mer). Maybe both rings are involved in B and only one ring in C???

Lines 419-420:"it remains unclear". Not really. Try Figure 6 DEF of Fei et al (PNAS 110: E2958–E2966). It's all about allostery!!

Line 480 and Fig 7C: "the magnitude of the effect (on PepQ folding) is proportional to the level of BSA-induced stimulation of the ATPase cycle". No it's not. A modest (1.5 fold) increase in the rate of ATPase yields a 4-fold increase in the rate of folding. There is something else going on here. Why doesn't the BSA compete with the PepQ??

Minor points:

Line 28: I don't like the use of the word "pull" since it implies the imposition of force, for which the authors have no evidence. This should be re-phrased.

Line 32: "reduced compactness of the PepQ monomer". I didn't find the cryoEM evidence for this compelling (cf Fig 4C and F). It's hardly mentioned in the rest of the text and I don't believe its omission would detract from the other arguments.

Line 107 and elsewhere: "pita-bread fold" Huh? Is that lab-jargon or legitimate. If the latter maybe we could have a reference?

Reviewer #2 (Remarks to the Author):

Chaperonin GroEL is an essential chaperone to assist protein folding in the cell. It has been known that GroEL accelerates the folding of some substrate proteins such as Rubisco. There has been much debate as to whether the mechanism of GroEL action is active or passive. The authors investigated to address this issue by using a homodimer enzyme, PepQ which is one of the obligate GroEL-dependent substrate proteins in E. coli, and found that GroEL actively stimulates folding of PepQ. Although the data presented here are technically sound, my overall impression is that this manuscript contains only an incremental extension of previous papers on the active role of GroEL in the assisted folding, by the authors and others for the following reasons. First, although the authors state that passive model is prevailing, active assistance of GroEL for the

folding of some proteins, such as Rubisco and DapA, has already been established. Then, the conclusion that GroEL actively assists the folding of PepQ is not novel but an extension of previous efforts including the work by the authors using Rubisco. Second, the data showing that slow spontaneous folding of PepQ is not caused by aggregation have already been demonstrated by some GroEL substrates, although there have been ongoing arguments on this issue. Third, the observation that C-terminal flexible region in GroEL has an essential role for the active assistance of GroEL, which is one of key findings based on both biochemical and cryo-EM structure, is not novel. The authors themselves have repeatedly shown the critical role of the C-terminal region in GroEL. Finally, in the Discussion section the authors examined whether the folding rate of PepQ is altered depending on the rate of GroEL ATPase cycle. The fact that repetitive cycling of substrate proteins by GroEL enhances productive folding has been known for a long time. Also, I note that utilization of PepQ to demonstrate the enhancement of folding rate is inappropriate if the folding rate is measured by the enzyme activity as in the case of this study, since PepQ is a homodimeric enzyme. Dimer formation after the quenching of GroEL ATPase, which took a 60 min in this study, would be too long to conclude that secondary effect, potentially leading to ambiguity, is negligible. Taken together, this manuscript does not contain a novel and important contribution to the understanding of the chaperonin GroEL-GroES for the standard of Nature Communications. Therefore, I don't recommend this manuscript to publish in the journal.

Reviewer #3 (Remarks to the Author):

In this work Weaver et al., suggest that GroEL actively stimulates folding of an endogenous protein substrate through interactions with the unstructured C terminal regions containing multiple glyglymet repeats. The work is sound in experimental scope, the data are well presented and is well written (i.e. easy for the readers to follow). Of particular interest is the demonstration that accelerated cycling leads to more rapid folding with tetradecameric GroEL vs the single ring versions. Furthermore, the cyro-EM work is offers a real convincing demonstration of the differences in capture complexes between the wild type (contains C terminal gly-gly-met repeats) and mutants lacking the c terminal tail. The differences in the protein substrate bound forms are striking and represent another clear example of the important previously ill-defined role that is played by the C terminal tail in folding proteins. The authors suggest that this partitioning and complex formation represents an active stimulation of folding by diminishing the population of kinetically slowed states that are momentarily trapped in incorrectly folded states. There are some minor points and corrections that need clarification and including expanded discussions of concerted chaperone systems.

1) The authors provide convincing evidence that the c terminal deletion does not result in similar fluorescent signals of transient folding species indicating that different populations result when C terminal tail interactions are intact. It is not clear that this folding intermediate species is independent or does the output signal and fluorescence increase is a result of the interaction of the folding protein with the numerous glyglymet tails. One cannot unequivocally state that this burst represents a separate folding species so the statement that this is a different folding interemediate has to be softened. One cannot rule out the possibility that it is the PepQ interaction with the chaperonin that gives rise to the observed burst (change) in fluorescence output.

2) Although this may seem to be a matter of semantics, describing the interaction with the C terminal tails as defining an active function may be a point of contention. For example, the authors include terms such as "pull the substrate in the active site" but the authors do not provide any data that includes any measurement of force to support the contention that this is an active process. For example, critics may argue that the C terminal tail regions would fit a passive capture system rather than one that would be involved in a more active involvement (direct physical involvement to unfold "kinetic trapped states". In some ways, the question here becomes a chicken and an egg problem. One has to ask, what comes first, the populated partially folded intermediate that "anneals" onto the chaperonin capture platforms (both in the apical domain and the C terminal tail region) or an as yet undefined physical force that pulls in or physically pulls apart regions that were prefolded in a trap folded state? Active reactions are more like the Physical

unfolding systems such as those associated with the ATPase function of the 26 S proteasome. This reviewer understands that forced unfolding from the standpoint of the changes in the apical domain but it is unclear how the c terminal tails are involved if at all in this "forced" unfolding since no forces were measured. This should be clarified for the readers.

3. Kinetic partitioning rather than thermodynamic arguments should be stressed in this paper. In addition, these solution conditions also do not resemble the cellular interior with respect to other chaperones influencing the initial capture species. The reaction described is a kinetically defined situation. Proteins with exactly the same folds (aka mutants) will shift from "non-stringent" to stringent based on the populations and properties of the partitioning folding intermediates so the key property at defining this event lies with the substrate and not with the chaperonin. As King and Sturtevant showed in 1992, the T_m of the final folds of the tailspike protein remained the same but the folding intermediate kinetic rates were affected by temperature partitioning (temperature sensitive folding mutants).

- Another case in point that is relevant for the E coli folding chaperone system is that the inclusion of the Hsp70/40/NEF may result in an entirely different set of capture complexes. As shown by the Bukau group, the Hsp70 system has the ability to bind many different extended forms and this interaction may alter the nature of the GroE capture population of the particular protein substrate in question. Would the E coli Hsp70 system also accelerate the apparent folding rates (unfolding misfolded traps) and what would a combination chaperone system (more like in vivo conditions) yield the same or different capture states under these low temperature conditions? Although these latter experiments are not required for this current submitted work, a discussion of the inclusion of more authentic E.coli chaperones is certainly in order. It was shown by Hartl's group that the stoichiometry requirement of the GroE chaperonin diminishes significantly when the Hsp70 chaperone system is included. If the chaperonin still captures the folding intermediates using the c terminal platform unfolding, this means that in some instances the release from the other chaperones still results in kinetically trapped intermediates.

- What would the distribution of bound species look like if the temperature was raised to 37C, the optimal temperature condition in E. coli? This is where upstream chaperone systems may play an important role in insuring that trapped species do not dominate the kinetic and thermodynamic folding energy landscape.

4. Although this group does not find extensive aggregation, the authors should offer an explanation as to what happened to the remaining population (~ 40%) of now inactive presumably refolded protein (only 60% yield). What happens to the inactive population? Where is the inactive protein? Does this protein exist as an inactive dimer? Trapped monomer? Merely answering this query with a statement that the levels are too small to detect and quantitate leaves this an open question. What level of dimer is detectable in the fluorescence correlation spectroscopy experiments? It would help if one could quantitate the amount active folded from inactive folded species. In various in vitro folding treatises and protocol reviews, Rainer Rudolph and Rainer Jaenicke also show that inclusion of low concentrations of non-denaturing chaotropes also facilitates folding albeit at slower rates than without the denaturant but folding yields increase. Are better refolding yields observed with spontaneous PepQ refolding when one includes a moderate amount of urea in the refolding solution? Again, this may have implications when one includes the other chaperone systems with the GroE system.

5. The authors use the term "crippled" to describe the double mutant maltose binding protein experiments. Since mutation plasticity is a hallmark of DNA encoded evolutionary events that appear to be aided by chaperones (See Tawfik's work examining directed protein evolution with GroE), it would be preferable to use another term here that would highlight the inherent drift in protein folding and partitioning onto the chaperone rather than call the protein crippled.

Response to Reviewer Comments

Reviewer 1:

Q.1 *The manuscript of Weaver et al addresses an important question with a large number of biophysical techniques with admirable rigor. For the most part I think their conclusions are justified by the experimental data (some exceptions to that below) so I conclude that it merits publication, subject to some revisions.*

A.1 We would like to thank the reviewer for the careful examination of the manuscript and many insightful comments, many of which have helped us improve the text. We also very much appreciate the strong and supportive comments about the value and impact of the work.

Q.2 *Lines 227-230: The argument outlined here is weak. If monomers don't form dimers at 2nM, why would they form polymers (aggregates)?*

A.2 We agree that the presentation of this argument is less than clear. We have now modified the text on p. 11 (lines 229-232) to provide a more complete explanation. The point made by the reviewer assumes that the stabilizing free energy for a PepQ aggregate cannot be any greater than that for the native dimer. However, there is no reason that this would have to be true. Indeed, it is very likely that the interaction surfaces and energies for a non-native aggregate are very different from the contacts made by the native dimer. While we cannot prove, for the moment, that this is so for PepQ, at a minimum it is a formal, and very reasonable, possibility that aggregates could form under conditions where the native dimer cannot. In fact, we and others have seen this exact behavior with another protein. At 25 °C and concentrations below ~ 20 nM, *R. rubrum* RuBisCO cannot readily form an active dimer in GroEL-mediated refolding assays ^{1,2}. However, based on both previous published estimates ¹, as well as our unpublished FRET, FCS and BAS data ³, RuBisCO aggregates quite readily form under these same conditions.

Q.3 *Figures 1 and 3: There is a discrepancy between the data in Fig 1B and the trp fluorescence data in Fig 3A and 3B. The t1/2 of spontaneous folding of PepQ is given as ~ 15 min in Fig 1B, But the t1/2 of the spontaneous trp fluorescence change is about 1 min. Conversely the t1/2 for the GroEL/ES/ATP folding is given as ~1 min in Fig 1B, but the t1/2 for the decrease in trp fluorescence is ~2min. Can the authors comment on this?*

A.3 We agree that the text could benefit from a better explanation for time scale differences between the Trp fluorescence changes and the rate of active dimer formation. We have added an additional explanation (pp. 12-13, lines 253-257) in order to clarify this issue.

We do not agree that this constitutes a significant discrepancy, however. It is clear that the changes we observe in the PepQ Trp fluorescence do not directly report on the rate limiting step(s) that constrain formation of the native state of the enzyme. In the case of spontaneous folding, the conformational transition that leads to the decrease in Trp fluorescence (Fig 3A) occurs much more rapidly than the rate at which the active dimer forms (Fig 1B). A simple

interpretation is that conformational changes of the monomer that affect the local environment around one or more Trp residues occurs significantly earlier than, and are not altered by, the subsequent rate-limiting transition that leads to the native dimer. At the same time, the observation that the decrease in Trp fluorescence does not occur when the PepQ monomer is maintained within the GroEL-GroES cavity (Fig 3C) suggests that the decrease in Trp fluorescence involves a structural transition that occurs only in free solution. Or that the decrease in Trp fluorescence is linked to transitions that occur within a nascent, but not fully folded, PepQ dimer. Because we do not have enough information about the detailed folding pathway of PepQ at this point, we feel that these ideas remain too speculative to warrant an extensive discussion in the text. More fundamentally, however, the mechanistic insights that we draw from the fluorescence measurements, which are primarily focused on changes in the conformational ensemble of the PepQ monomer inside the GroEL cavity, do not require that they be directly linked to the committed step.

Finally, the rate of the Trp fluorescence change in the presence of GroEL-GroES is not slower than the rate of enzymatically active dimer formation, as the reviewer suggests. The $t_{1/2}$ for GroEL-mediated dimer formation must be slower than the folding of the monomer, which (from Fig 1B) is 1.1 min, while the $t_{1/2}$ for the falling phase of the Trp fluorescence change in the presence of GroEL-GroES in Fig 3B is 0.8 min. To eliminate potential confusion on this point, we have now added the explicit rate constants for the fits of the Trp fluorescence data to the legend of Fig 3.

Q.4 *Figure 3B and 3C: The data has been normalized. Are the real amplitudes the same for 3B (a 14 mer) and 3C (a 7mer). Maybe both rings are involved in B and only one ring in C???*

A.4 The absolute fluorescence amplitudes for the tetradecamer and SR1 experiments in Fig. 3 differ by only ~ 10%. Consequently, there is little, if any, information contained in the magnitude of the fluorescence amplitudes. In order to present a more clear comparison of the kinetic changes, however, we have chosen to normalize these fluorescence data. The relatively minor difference in fluorescence amplitude strongly suggests that there is no significant contribution from the second tetradecamer ring of the GroEL oligomer in these experiments. Additionally, in both cases, the chaperonin is present in excess (~ 2x for GroEL and ~ 3x for SR1) relative to the PepQ monomer. Thus, the vast majority of the GroEL tetradecamers are only ever associated with a single PepQ monomer during this experiment.

Q.5 *Lines 419-420:"it remains unclear". Not really. Try Figure 6 DEF of Fei et al (PNAS 110: E2958–E2966). It's all about allostery!!*

A.5 We certainly agree with the reviewer that the fundamental nature of the signaling between a substrate engaged GroEL ring and ejection of GroES from the opposite ring must be deeply entwined in the allostery of the GroEL machine. And while the elegant work in Fei et al. does present some strong arguments for allosteric signaling pathways in an ADP-bound GroEL ring, this structural analysis does not directly address the point we are making here: namely, how the binding of a non-native substrate protein is structurally linked to ejection of GroES from the opposite ring. Because neither GroES nor non-native substrate protein were present in the

crystal structures that are the basis of the study in Fei et al., the specific transitions linking substrate protein binding to GroES release remain to be determined in detail.

Q.6 *Line 480 and Fig 7C: “the magnitude of the effect (on PepQ folding) is proportional to the level of BSA-induced stimulation of the ATPase cycle”. No it’s not. A modest (1.5 fold) increase in the rate of ATPase yields a 4-fold increase in the rate of folding. There is something else going on here. Why doesn’t the BSA compete with the PepQ??*

A.6: Our claim here was never that the change in ATPase rate yielded an “equivalent” response in folding rate. Our point was that a positive increase in ATPase rate induced a positive increase in PepQ folding rate, whatever the exact magnitude of the proportionality. To try and alleviate the erroneous impression caused by our use of the word “proportional,” we have removed this word from the text on p. 23, line 494. We have now re-written this sentence to present a softer claim, namely that the increase in folding rate “mirrors” the increase in ATPase rate.

Importantly, our previous work on the linkage between the GroELS ATPase rate and stimulated folding of RuBisCO⁴ found a similar mirroring. In this work, we examined the stimulated folding of RuBisCO by the cycling GroELS system as a function of temperature. We observed an ~40% increase in the observed steady state rate of ATP hydrolysis by GroELS when the temperature was increased from 25 °C to 35 °C. This increase in ATP turnover, however, resulted in a 2.5- to 3-fold increase in the observed rate of RuBisCO folding. Thus, for both PepQ and RuBisCO, a relatively “modest” increase in ATP turnover results in an “amplified” facilitated folding response. We have now added a more explicit description of these prior observations to the discussion on pp. 24-25, lines 495-500.

We believe that BSA does, in fact, compete with the PepQ, but simply not very well. We note that at concentrations of BSA that have only a small impact on the steady state ATPase rate of the GroEL-ES system (0.1 mg/ml BSA or lower; Figure 7 B and C), additions of BSA suppress the observed folding rate of PepQ. This observation is most simply interpreted as being due to BSA competing with PepQ binding to GroEL under conditions where the chaperonin provides no more folding stimulation than in the absence of BSA. At higher BSA concentrations, however, the extent of the amplified stimulation provided, as a result of accelerated ATP turnover, begins to exceed the impact of the weak competition between the BSA and PepQ.

Q.7 *Line 28: I don’t like the use of the word “pull” since it implies the imposition of force, for which the authors have no evidence. This should be re-phrased*

A.7 In order to reduce potential confusion about how the C-termini are impacting the PepQ folding intermediate, we have removed the word “pull” from the manuscript. However, we remain convinced that the C-termini are, in fact, impacting the PepQ folding intermediate in a way that results in the directed transport of the monomer toward the bottom of the GroEL cavity. The third reviewer raised a similar concern and we have responded in greater detail below.

Q.8 Line 32: *:"reduced compactness of the PepQ monomer". I didn't find the cryoEM evidence for this compelling (cf Fig 4C and F). It's hardly mentioned in the rest of the text and I don't believe its omission would detract from the other arguments.*

A.8 We agree that our presentation of the structural data in support of this argument was less than ideal. To address this ambiguity, we have measured the observed PepQ density visible in the cryo-EM maps of both the $\Delta 526$ and the wild type GroEL complexes at matched contour levels. When the GroEL subunits, (shown in Fig. 6 C and D), are isosurfaced to the same volume size of $\sim 61,000 \text{ \AA}^3$, the observed volumes for the PepQ monomer measures $8,564 \text{ \AA}^3$ in the $\Delta 526$ complex and $2,696 \text{ \AA}^3$ in the wild type GroEL complex. This observation is highly consistent with our conclusion that the PepQ monomer is much more unfolded in the wild type GroEL, leading to its weaker density in the cryo-EM reconstruction and much greater protease sensitivity (Fig. 3F). We have revised both Fig.6 and the manuscript text to include these observations.

Q.9 Line 107 and elsewhere: *"pita-bread fold" Huh? Is that lab-jargon or legitimate. If the latter maybe we could have a reference?*

A.9. This is a recognized fold for the superfamily of metalloproteases to which PepQ belongs⁵⁻⁸. We neglected to include some of these references in key places in the text. We have now rectified these omissions.

Reviewer 2:

Q.1 *Although the data presented here are technically sound, my overall impression is that this manuscript contains only an incremental extension of previous papers on the active role of GroEL in the assisted folding, by the authors and others for the following reasons. First, although the authors state that passive model is prevailing, active assistance of GroEL for the folding of some proteins, such as Rubisco and DapA, has already been established. Then, the conclusion that GroEL actively assists the folding of PepQ is not novel but an extension of previous efforts including the work by the authors using Rubisco.*

A.1 We respectfully, and strongly, disagree. As we point out in the manuscript, there are fundamentally serious problems with trying to derive general conclusions about accelerated protein folding by chaperonins from only two, and in some cases, contradictory examples. At the same time, the existence and importance of active folding remains controversial (for example, see^{9,10}). Our studies are, demonstrably, the first to show such an effect with this protein or any member of this family of protein folds. Given the substantial gaps in our current knowledge about chaperonin-facilitated protein folding, simply knowing that a protein is a GroEL substrate does not, in any way, demonstrate that it is actively folded by GroEL. The fundamental point of our study was to show that, with a biologically matched substrate protein from a fold family that had never been studied before, the signatures of active protein folding by GroEL can be robustly observed.

Q.2 *Second, the data showing that slow spontaneous folding of PepQ is not caused by aggregation have already been demonstrated by some GroEL substrates, although there have been ongoing arguments on this issue.*

A.2 Again, because this point has been made, and disputed, for a couple of other proteins, does not in any way demonstrate that this is a settled issue. Quite the contrary. While we are convinced, in large part now because of the novel data we present for PepQ, that many stringent GroEL substrate proteins are likely to behave similarly to RuBisCO and PepQ, a general conclusion of this type, as is so often true in biology, cannot be robustly drawn without multiple clean examples. This work, in part, now provides that convincing additional example with a protein from a distinct fold family.

Q.3 *Third, the observation that C-terminal flexible region in GroEL has an essential role for the active assistance of GroEL, which is one of key findings based on both biochemical and cryo-EM structure, is not novel. The authors themselves have repeatedly shown the critical role of the C-terminal region in GroEL.*

A.3 The point of the manuscript is not to simply show that the C-termini are important. As the reviewer notes, this has been shown by us and others before. However, it is also worth pointing out that there is controversy on even this point^{11,12}. More importantly, the point of our work in the current manuscript is to provide direct and specific insight into how the C-termini impact the mechanism of GroEL-mediated folding. In particular, the demonstration that the C-termini make physical contact with a folding intermediate, help retain the folding intermediate inside the cavity prior to the binding of nucleotide and participate in partial unfolding of the substrate protein. We agree that some elements of these conclusions have been suggested in prior work, including our own. By and large, however, these previous studies involved indirect observation or inference. Here, we present direct physical evidence for these conclusions in the form of completely novel, sub-nanometer cryo-EM structures. The conclusions we can draw from these structures, in combination with the other solution data we present, provide uniquely powerful insight into the mechanism of GroEL-mediated protein folding and the role of the C-termini.

Q.4 *Finally, in the Discussion section the authors examined whether the folding rate of PepQ is altered depending on the rate of GroEL ATPase cycle. The fact that repetitive cycling of substrate proteins by GroEL enhances productive folding has been known for a long time.*

A.4 While suggested by initial work from Lorimer and colleagues, and supported by some of our previous observations, the idea that repetitive cycling stimulates folding is strongly disputed by other major groups in the field (principally Hartl and colleagues; see⁹ for example). Our observations here with PepQ strike at the very heart of this deep controversy over the relative importance of repetitive unfolding and enhanced folding. While our earlier observations on RuBisCO provided, we would argue, some of the most convincing experimental data in support of this idea, a single observation on a non-homologous substrate protein does not, once again, yield a general mechanism. The entire point of these observations is to help provide the evidence for that general conclusion.

Q.5 Also, I note that utilization of PepQ to demonstrate the enhancement of folding rate is inappropriate if the folding rate is measured by the enzyme activity as in the case of this study, since PepQ is a homodimeric enzyme. Dimer formation after the quenching of GroEL ATPase, which took a 60 min in this study, would be too long to conclude that secondary effect, potentially leading to ambiguity, is negligible.

A.5 We believe the assumption behind this comment is flawed. The fact that the final stages of PepQ dimer formation can be slow, especially at low protein concentrations, is not relevant to the key mechanistic argument presented. The important limiting event is the rate of commitment, or the rate at which the substrate protein populates a conformation that no longer requires any action of the chaperonin. It is this step or steps that GroEL impacts. If subsequent steps that lead to, or finalize, enzymatically active oligomer formation are slow, it simply does not matter for understanding the role that a molecular chaperone plays in the earlier steps which limit access to the committed state. These slower steps are independent of the chaperone and therefore not relevant to understanding chaperone-facilitated protein folding. Consequently, the rate at which PepQ acquires its enzymatically active state in the presence or absence of GroEL does, actually, accurately report on the impact that the chaperonin has on the rate-limiting committed step(s).

It is worth noting that most of the highly dependent GroEL substrate proteins that have been studied in detail are dimers or oligomers, including RuBisCO, DapA, and MDH. Well-established folding assays based on the re-gain of enzymatic activity have been used in all of these cases and have been successfully and quantitatively applied to the study of GroEL^{13,14}. In situations where very low protein concentrations are employed, it is not at all unusual for these assays to involve extended incubations of folding reactions to permit active oligomer assembly^{2,13,14}.

Reviewer 3:

Q.1 In this work Weaver et al., suggest that GroEL actively stimulates folding of an endogenous protein substrate through interactions with the unstructured C terminal regions containing multiple glyglymet repeats. The work is sound in experimental scope, the data are well presented and is well written (i.e. easy for the readers to follow). Of particular interest is the demonstration that accelerated cycling leads to more rapid folding with tetradecameric GroEL vs the single ring versions. Furthermore, the cyro-EM work is offers a real convincing demonstration of the differences in capture complexes between the wild type (contains C terminal gly-gly-met repeats) and mutants lacking the c terminal tail.

A.1 We would also like to thank this reviewer for their detailed examination of our manuscript. Many of the issues raised helped us to improve the work. We also very much appreciate the strong and supportive comments about the value and impact of the work.

Q.2 The authors provide convincing evidence that the c terminal deletion does not result in similar fluorescent signals of transient folding species indicating that different populations result when C terminal tail interactions are intact. It is not clear that this folding intermediate species is independent or does the output signal and fluorescence

increase is a result of the interaction of the folding protein with the numerous glyglymet tails. One cannot unequivocally state that this burst represents a separate folding species so the statement that this is a different folding interemediate has to be softened. One cannot rule out the possibility that it is the PepQ interaction with the chaperonin that gives rise to the observed burst (change) in fluorescence output.

A.2 We agree that the observed fluorescence burst could, in principle, be explained in two ways. As we have suggested, the burst could come from a distinct ensemble of PepQ folding intermediates that are populated upon GroES binding in the presence of the C-termini, but not in their absence. This intermediate would then possess an average Trp quantum yield that is very different from that of the PepQ monomer in the absence of the C-termini. Alternately, the PepQ folding intermediate could be conformationally identical in the presence and absence of the C-termini, but a physical interaction between the C-termini and the PepQ Trp residues could result in a direct quantum yield perturbation of a subset of the Trp residues. Upon removal of the C-termini, this direct alteration of the Trp fluorescence would be released.

In order to avoid potential confusion, we have softened the language on this point in the Abstract and Results sections, shifting from stating that the Trp fluorescence data “demonstrates” a change in folding pathway to “suggests” or “is consistent with.” Even so, we feel that it is far more likely that the C-termini do, in fact, alter the conformational ensemble of the PepQ monomer that is populated immediately following GroES binding, a shift that is associated with a change in the average quantum yield of the Trp residues. We note that:

1. A model in which the changes in Trp fluorescence result only from direct contact (quenching or enhancement) of the PepQ Trp residues by the C-termini makes a strong prediciton: the level of Trp fluorescence produced from a PepQ folding intermediate bound to a full length GroEL ring (i.e. one with the C-termini intact) compared to a $\Delta 526$ ring should be very different. Specifically, when the tails are removed, this model predicts that the observed Trp fluorescence should be far higher (in the case of a quenching effect) or lower (in the case of an enhancing effect) than when they are present. However, this is not the case. As shown in Fig 3E, the level of Trp fluorescence at the beginning of this stopped flow experiment, in which non-native PepQ is bound to either an SR1 or SR $\Delta 526$ ring is essentially the same for both GroEL variants. In the case of the SR1 ring, this represents a non-native PepQ folding intermediate in intimate contact with the C-termini. This observation is not consistent with simple direct quenching or enhancement of the PepQ Trp fluorescence by the C-termini.
2. Our argument for a distinct spectrum of conformational states between a full length and a $\Delta 526$ ring at the start of, or just after, encapsulation does not rely exclusively on the Trp fluorescence data. The cyro-EM and proteolysis also strongly suggest that this is likely to be the case.

Q.3 *Although this may seem to be a matter of semantics, describing the interaction with the C terminal tails as defining an active function may be a point of contention. For example, the authors include terms such as “pull the substrate in the active site” but the authors do not provide any data that includes any measurement of force to support the contention that this is an active process. For example, critics may argue that the C terminal tail regions would fit a passive capture system rather than one that would be*

involved in a more active involvement (direct physical involvement to unfold “kinetic trapped states”. In some ways, the question here becomes a chicken and an egg problem. One has to ask, what comes first, the populated partially folded intermediate that “anneals” onto the chaperonin capture platforms (both in the apical domain and the C terminal tail region) or an as yet undefined physical force that pulls in or physically pulls apart regions that were prefolded in a trap folded state? Active reactions are more like the Physical unfolding systems such as those associated with the ATPase function of the 26 S proteasome. This reviewer understands that forced unfolding from the standpoint of the changes in the apical domain but it is unclear how the c terminal tails are involved if at all in this “forced” unfolding since no forces were measured. This should be clarified for the readers.

The issue raised here with respect to the term “pull” is similar to that expressed by the first reviewer. As we noted above, we have removed this word from the manuscript and softened the language we use concerning this point. However, we remain convinced that the C-termini are, in fact, impacting the PepQ folding intermediate in a way that results in the transport of the monomer toward the bottom of the GroEL cavity. It is important to note that a pulling action does not require direct linkage to a local power source like ATP hydrolysis in order for this to be an active process. There are several points to address in our response to this comment and we enumerate them below:

1. We recognize that we did not conduct any explicit force measurements and that we do not have the data to perform a detailed force analysis. We also agree with the reviewer(s) that this limits how specific we can be in our statements about the impact of the C-termini. We feel that the placement of the C-termini and cryo-EM data are most consistent with a model in which a significant force is applied to the substrate protein that is directed through the location of the C-termini at the bottom of the cavity. However, because there are other regions of contact between the GroEL cavity and the PepQ substrate, we cannot rule out a more complicated force model where no net force is applied to the substrate through the C-termini while the substrate undergoes both expansion and a net displacement of its center of mass largely toward the C-termini.
2. With respect to the question of whether the “passive” binding of the C-termini could, even in principle, impose a force sufficient to pull or unfold the substrate protein, we consider the example of protein secretion across membranes. This is a process where an inherently passive event (i.e. simple binding) does work on, and therefore applies a directional force to, a protein. The Rapoport group (and others) have demonstrated that a simple binding ratchet, driven by the asymmetric distribution of a binding factor on one side of a membrane, is not only sufficient to drive directional transport, but can apply a pulling force powerful enough to cause some marginally stable proteins to unfold ^{15,16}. While a molecular chaperone of the Hsp70 family is usually the binding agent that powers this transport ratchet, ATP hydrolysis is primarily used to control binding and release of the chaperone from the target protein. The Hsp70 does not appear to impart direct mechanical pulling. Rather, the tight association of the chaperone with the transport protein on one side of the membrane results in a directional diffusive bias of the protein’s movement into the compartment with the chaperone. Importantly, the same directional transport can be accomplished with nothing more than antibodies directed at a poly-epitope placed at one

end of the transport protein ¹⁶. Thus, a purely passive binding process can, in fact, be rectified into the directional application of force to a polypeptide.

3. The question about how the C-termini are involved in forced unfolding is a good one. We first note that forced unfolding is only one of two unfolding phases imposed by GroEL on a folding intermediate. The binding-driven unfolding that we first characterized with RuBisCO ^{4,17-19}, examine further here with PepQ, and that has now been observed by NMR ²⁰, occurs during and after initial capture of the folding intermediate (prior to ATP binding). In our prior work on RuBisCO, we suggested that the C-termini might provide a secondary binding platform for a folding intermediate at the base of the GroEL cavity ^{17,21}. Our work here with PepQ now directly confirms this supposition. At the same time, we also showed that the binding of ATP to a RuBisCO-occupied GroEL ring results in a very rapid, forced unfolding event ^{4,17,18}. As with binding-driven unfolding, forced unfolding is substantially attenuated when the C-termini are removed ¹⁷. So, the simple model we have proposed is that the C-termini provide an anchor point at the bottom of the cavity that holds or restrains the folding intermediate toward the bottom of the cavity as the apical domains move following ATP binding ^{17,21}. Because we do not yet possess the tools needed to examine forced unfolding with PepQ, we have focused on binding-associated unfolding in this manuscript on PepQ. As requested by the reviewer, we have now added two additional paragraphs to the the Discussion section (pp. 24-25, lines 502-534) to clarify these issues.

Q.4 *Kinetic partitioning rather than thermodynamic arguments should be stressed in this paper. In addition, these solution conditions also do not resemble the cellular interior with respect to other chaperones influencing the initial capture species. The reaction described is a kinetically defined situation. Proteins with exactly the same folds (aka mutants) will shift from "non-stringent" to stringent based on the populations and properties of the partitioning folding intermediates so the key property at defining this event lies with the substrate and not with the chaperonin. As King and Sturtevant showed in 1992, the T_m of the final folds of the tailspike protein remained the same but the folding intermediate kinetic rates were affected by temperature partitioning (temperature sensitive folding mutants).*

A.4 We agree that kinetic partitioning is an absolutely key concept for understanding how molecular chaperones function. However, we are uncertain as to the reviewer's point about thermodynamic arguments. The foundational arguments we present around the mechanisms of GroEL in facilitated protein folding of PepQ are explicitly kinetic. We refer to the "kinetically trapped" PepQ intermediate throughout the text and our discussion of GroEL mechanisms is focused on the idea that GroEL resolves this fundamentally kinetic problem. In order to further emphasize this point, however, we have re-written the first two sentences of the Discussion section to emphasize the idea of kinetic partitioning and the fact that chaperonins are fundamentally kinetic editors of protein folding reactions.

While we certainly have developed and exploited a "kinetically defined" situation in these *in vitro* studies, we fundamentally do not see this as a significant issue. Nor is, we would argue, the fact that the conditions we employ are not identical to cellular conditions. Virtually any well defined and well controlled *in vitro* mechanistic study is subject to exactly the same limitations. We make no claim that the behavior we observe fully explains all possible ways that a facilitated

folding reaction by GroEL could occur when considering the far more complex solution conditions in a living cell. However, the power and essential correctness of the mechanistic insights derived from well-defined, *in vitro* experiments seem, to us at least, exceedingly well established.

We agree that the term “stringent” should refer to a property of the individual substrate protein and not a given molecular chaperone system. This is the exact sense in which we use this word in the manuscript. In the most general terms, a “stringent” GroEL substrate protein is one that depends upon the full GroELS system to fold efficiently (i.e. GroEL alone is insufficient) and that either does not fold at all spontaneously, or at least folds very poorly. Truly stringent substrate proteins are also those for which other molecular chaperone systems cannot be substituted for GroELS; these proteins have an obligate dependence on GroELS for efficient and rapid folding. While these definitions were originally operational, several elegant proteomic studies of GroEL-dependent proteins in *E. coli* have provided an excellent *in vivo* correlate of these definitions^{22,23}. Nonetheless, it remains true that stringency is still somewhat dependent on the solution or environmental conditions one employs (or occurrence of mutations). However, we tend to employ an even more strict usage of this term, reserving it for proteins that display a particular type of kinetic behavior. Namely, we view truly stringent proteins are those whose spontaneous folding cannot be rescued by simple dilution to very low protein concentrations (e.g. RuBisCO, PepQ and DapA). Thus our usage of “stringent” is generally reserved for proteins that, in addition to being highly dependent on the full GroELS system both *in vivo* and *in vitro*, also display refractory spontaneous folding even under solution conditions where aggregation does not occur.

Q.5 *Another case in point that is relevant for the E coli folding chaperone system is that the inclusion of the Hsp70/40/NEF may result in an entirely different set of capture complexes. As shown by the Bukau group, the Hsp70 system has to ability to bind many different extended forms and this interaction may alter the nature of the GroE capture population of the particular protein substrate in question. Would the E coli Hsp70 system also accelerate the apparent folding rates (unfolding misfolded traps) and what would a combination chaperone system (more like in vivo conditions) yield the same or different capture states under these low temperature conditions? Although these latter experiments are not required for this current submitted work, a discussion of the inclusion of more authentic E.coli chaperones is certainly in order. It was shown by Hartl’s group that the stoichiometry requirement of the GroE chaperonin diminishes significantly when the Hsp70 chaperone system is included. If the chaperonin still captures the folding intermediates using the c terminal platform unfolding, this means that in some instances the release from the other chaperones still results in kinetically trapped intermediates.*

A.5 We appreciate this reviewer’s enthusiasm about the role of other chaperone systems and the possibility of the inclusion of the Hsp70 system in our PepQ experiments. We agree that these are very interesting questions and we are actively pursuing them. However, we feel these studies are beyond the scope of the current work. Nonetheless, we have briefly expanded the end of the Discussion section (p. 27, lines 564-569) in order to recognize this point.

Q.6 *What would the distribution of bound species look like if the temperature was raised to 37C, the optimal temperature condition in E. coli? This is where upstream chaperone systems may play an important role in insuring that trapped species do not dominate the kinetic and thermodynamic folding energy landscape.*

A.6 This is also an excellent point that will be explored in later work. Ideally, we need to develop more detailed probes for PepQ (e.g. intra-molecular FRET pairs at a minimum) in order to execute this experiment correctly. These new probes are in development, but not currently in hand.

Q.7 *Although this group does not find extensive aggregation, the authors should offer an explanation as to what happened to the remaining population (~ 40%) of now inactive presumably refolded protein (only 60% yield). What happens to the inactive population? Where is the inactive protein? Does this protein exist as an inactive dimer? Trapped monomer? Merely answering this query with a statement that the levels are too small to detect and quantitate leaves this an open question. What level of dimer is detectable in the fluorescence correlation spectroscopy experiments? It would help if one could quantitate the amount active folded from inactive folded species. In various in vitro folding treatises and protocol reviews, Rainer Rudolph and Rainer Jaenicke also show that inclusion of low concentrations of non-denaturing chaotropes also facilitates folding albeit at slower rates than without the denaturant but folding yields increase. Are better refolding yields observed with spontaneous PepQ refolding when one includes a moderate amount of urea in the refolding solution? Again, this may have implications when one includes the other chaperone systems with the GroE system.*

A1: The question about the fate of PepQ monomers that do not reach the native state is a good one. At least at low protein concentrations, the data we present argues very strongly that this PepQ subpopulation remains in solution as a kinetically trapped monomer. We have now added a new sentence making this explicit point at the end of the Results section where the single molecule coincidence experiments are described (p. 12, lines 242-244).

Whether PepQ can also form a non-native, inactive dimer is an interesting question. It is formally possible that a subpopulation of non-native PepQ dimers could be present in our FCS experiments. However, the ability to detect this subpopulation by FCS would depend on several factors which are difficult to explicitly measure here, including the actual difference in mean diffusion time of a committed monomer compared to a hypothetical non-native dimer and the relative population fractions. Even under ideal conditions, robustly observing such a dimer subpopulation in a mixed sample by FCS would be challenging.

It is worth considering, however, a strong prediction that follows from the assumption that non-native dimers form during the FCS experiments. Because the native dimer cannot form under the conditions of this experiment (2 nM monomer), a postulated non-native dimer would have to possess a far higher stability and far larger bimolecular binding rate constant than the native dimer. If this were the case, a substantial population of hyper-stable, non-native dimers would have been readily detected in the single molecule co-incidence experiment. Yet, none were (Fig. 3D). We would therefore argue that at least under the low protein concentrations used in the FCS experiments, the vast majority of the PepQ protein remains monomeric.

Even with these observations, it remains formally possible that non-native dimers might form at higher protein concentrations. Overall, our data suggest that if non-native dimers do form at higher PepQ concentrations, they are either not extensively populated at concentrations up to ~ 100 nM, or their formation has only a secondary impact. The response of PepQ refolding curves to changes in protein concentration (Fig. 1D), in combination with the lack of any significant signal in inter-molecular FRET experiments (Fig S2C), indicates that non-native dimer formation is not a dominant process. In addition, we have examined spontaneous PepQ folding using native gel electrophoresis (Response Figure 1). We find that this assay robustly detects formation of the native PepQ dimer. Importantly, the intensity of the observed dimer band in the

native gel electrophoresis experiment, when converted to fraction of input PepQ, recapitulates the refolding yield measured by enzymatic activity in Fig 1D. However, if the ~ 40-50% of the PepQ that did not spontaneously reach the native state was trapped as a non-native dimer, we would expect to observe either a second, discreet band or find that the amplitude of the observed dimer band was much greater than the amplitude of the activity-based refolding assay (likely approaching 100%). Neither of these behaviors was observed. These results also argue that, at least up to ~ 100 nM monomer concentrations, the kinetically trapped PepQ remains essentially monomeric. Because we felt the fate of the non-native, trapped PepQ fraction in spontaneous refolding experiments was sufficiently explained by the other data we present, we have chosen not to include these native gel experiments in the manuscript.

Response Figure 1. Examination of PepQ folding by native gel electrophoresis. (A) Samples of denatured PepQ-24F (100 nM) were subjected to either spontaneous (lanes 4-6) or GroEL-mediated (lanes 7-9) folding and then examined by native gel electrophoresis. Lanes 1-3 are a set of loading controls of native PepQ used to calibrate the amount of folded PepQ observed following fluorescence imaging of the gel. The position of the native PepQ dimer and the sample well are indicated. A faint smear at the base of the sample well observed during spontaneous folding suggests the presence of a heterogeneous population of PepQ monomer that does not migrate as a discreet species by native electrophoresis. (B) The observed integrated intensities of the native PepQ dimer band, converted to fraction of total input, are shown.

Q.8 *The authors use the term “crippled” to describe the double mutant maltose binding protein experiments. Since mutation plasticity is a hallmark of DNA encoded evolutionary events that appear to be aided by chaperones (See Tawfik’s work examining directed protein evolution with GroE), it would be preferable to use another term here that would highlight the inherent drift in protein folding and partitioning onto the chaperone rather than call the protein crippled.*

In order to eliminate any potentially pejorative implications of the word “crippled,” we have removed this term. While we agree that there are very important implications to how molecular chaperones impact protein evolution through “phenotypic buffering,” this is a topic well outside the subject of the current manuscript. At the same time, we feel that it is important to point out that the double mutant MBP employed in prior studies is not a naturally occurring variant of this protein. This is a particularly relevant point for *E. coli* MBP, which is a well-studied secretory protein. Because secretory proteins do not, in general, ever come into contact with the Hsp60 system, it is exceedingly unlikely that even evolutionarily relevant mutations that alter this protein’s stability or folding would be buffered by GroEL. Thus, the argument about whether one can derive biological relevance from *in vitro* experiments conducted with heterologous substrate proteins or non-natural mutants of otherwise GroEL-independent proteins cannot be cleanly dismissed without convincing examples. Our point here was, in part, to provide such a rigorous example.

References

1. van der Vies, S. M., Viitanen, P. V., Gatenby, A. A., Lorimer, G. H. & Jaenicke, R. Conformational states of ribulosebiphosphate carboxylase and their interaction with chaperonin 60. *Biochemistry* **31**, 3635–3644 (1992).
2. Brinker, A. *et al.* Dual function of protein confinement in chaperonin-assisted protein folding. *Cell* **107**, 223–33. (2001).
3. Puchalla, J., Krantz, K., Austin, R. & Rye, H. S. Burst analysis spectroscopy: a versatile single-particle approach for studying distributions of protein aggregates and fluorescent assemblies. *Proceedings of the National Academy of Sciences of the United States of America* **105**, 14400–14405 (2008).
4. Lin, Z., Puchalla, J., Shoup, D. & Rye, H. S. Repetitive Protein Unfolding by the trans Ring of the GroEL-GroES Chaperonin Complex Stimulates Folding. *Journal of Biological Chemistry* **288**, 30944–30955 (2013).
5. Park, M.-S. *et al.* Catalytic properties of the PepQ prolidase from Escherichia coli. *Arch Biochem Biophys* **429**, 224–230 (2004).
6. Weaver, J., Watts, T., Li, P. & Rye, H. S. Structural basis of substrate selectivity of *E. coli* prolidase. *PLoS ONE* **9**, e111531 (2014).
7. Bazan, J. F., Weaver, L. H., Roderick, S. L., Huber, R. & Matthews, B. W. Sequence and

- structure comparison suggest that methionine aminopeptidase, prolidase, aminopeptidase P, and creatinase share a common fold. *Proc Natl Acad Sci U S A* **91**, 2473–2477 (1994).
8. Lowther, W. T. & Matthews, B. W. Structure and function of the methionine aminopeptidases. *Biochim Biophys Acta* **1477**, 157–167 (2000).
 9. Hayer-Hartl, M., Bracher, A. & Hartl, F. U. The GroEL-GroES Chaperonin Machine: A Nano-Cage for Protein Folding. *Trends Biochem Sci* **41**, 62–76 (2016).
 10. Horwich, A. L. & Fenton, W. A. Chaperonin-mediated protein folding: using a central cavity to kinetically assist polypeptide chain folding. **42**, 83–116 (2009).
 11. Burnett, B. P., Horwich, A. L. & Low, K. B. A carboxy-terminal deletion impairs the assembly of GroEL and confers a pleiotropic phenotype in *Escherichia coli* K-12. *J Bacteriol* **176**, 6980–6985 (1994).
 12. McLennan, N. F., Girshovich, A. S., Lissin, N. M., Charters, Y. & Masters, M. The strongly conserved carboxyl-terminus glycine-methionine motif of the *Escherichia coli* GroEL chaperonin is dispensable. *Mol Microbiol* **7**, 49–58 (1993).
 13. Goloubinoff, P., Christeller, J. T., Gatenby, A. A. & Lorimer, G. H. Reconstitution of active dimeric ribulose biphosphate carboxylase from an unfolded state depends on two chaperonin proteins and Mg-ATP. *Nature* **342**, 884–889 (1989).
 14. Georgescauld, F. *et al.* GroEL/ES chaperonin modulates the mechanism and accelerates the rate of TIM-barrel domain folding. *Cell* **157**, 922–934 (2014).
 15. Liebermeister, W., Rapoport, T. A. & Heinrich, R. Ratcheting in post-translational protein translocation: a mathematical model. *Journal of Molecular Biology* **305**, 643–656 (2001).
 16. Matlack, K. E., Misselwitz, B., Plath, K. & Rapoport, T. A. BiP acts as a molecular ratchet during posttranslational transport of prepro- α factor across the ER membrane. *Cell* **97**, 553–564 (1999).
 17. Weaver, J. & Rye, H. S. The C-terminal Tails of the Bacterial Chaperonin GroEL Stimulate Protein Folding by Directly Altering the Conformation of a Substrate Protein. *Journal of Biological Chemistry* **289**, 23219–23232 (2014).
 18. Lin, Z., Madan, D. & Rye, H. S. GroEL stimulates protein folding through forced unfolding. *Nat Struct Mol Biol* **15**, 303–311 (2008).
 19. Lin, Z. & Rye, H. S. Expansion and compression of a protein folding intermediate by GroEL. *Mol Cell* **16**, 23–34 (2004).
 20. Libich, D. S., Tugarinov, V. & Clore, G. M. Intrinsic unfoldase/foldase activity of the chaperonin GroEL directly demonstrated using multinuclear relaxation-based NMR. *Proceedings of the National Academy of Sciences of the United States of America* **112**, 8817–8823 (2015).

21. Chen, D.-H. *et al.* Visualizing GroEL/ES in the Act of Encapsulating a Folding Protein. *Cell* **153**, 1354–1365 (2013).
22. Kerner, M. *et al.* Proteome-wide analysis of chaperonin-dependent protein folding in *Escherichia coli*. *Cell* **122**, 209–220 (2005).
23. Fujiwara, K., Ishihama, Y., Nakahigashi, K., Soga, T. & Taguchi, H. A systematic survey of in vivo obligate chaperonin-dependent substrates. *EMBO J* **29**, 1552–1564 (2010).

REVIEWERS' COMMENTS:

Reviewer #1 (Remarks to the Author):

I appreciate the extensive responses of the authors to my previous comments as well as to those of the other reviewers. I am satisfied with their explanatory comments - the manuscript has been improved. I recommend publication

Reviewer #3 (Remarks to the Author):

IN this resubmission, the authors adequately address and expand on the comments provided by this reviewer (reviewer III).

Very Small additions may be necessary in the discussion and results sections.... Publication with very minor additions see below

a) The authors correctly acknowledge that the trp fluorescence could also be enhanced by the interaction with the C terminal tails. They show that this is not the case in the comparison instances C terminal removal vs wild type trp fluorescence.

b) minor point - C termini involved in pulling. Removal of pulling is agreed upon. However, there is still a matter of small disagreement with respect to an active vs passive mechanism. They make the argument using the hsp70 capture and prevention of back diffusion for a brownian ratchet type mechanism. This is not quite the same mechanism that is used GroEL because the Rapoport reference transport system is dependent on a leading strand protein unfolding from a threading perspective rather than a capture of an entire partially folded protein. In that there may be some further annealing of GGM repeats interacting with the partially unfolded protein, the actions may still be passive because the initial capture (for example initially at the interior rim) is a dynamic equilibrium process where simultaneous interaction of the substrate with the rim and end C termini GGM does not result in kinetically trapped complex at the top of the rim but rather, the initially bound substrate (perhaps represented by the state of the C terminal depleted species - may actually represent the initial trapped form)) has the opportunity to dissociate from the rim region (momentarily) and then further partition with or interacts with the tighter binding GGM tails. Based on H. Saibil's work, there certainly may be more intermediate GroEL complexes that have not been adequately resolved (i.e. some substrate on the top, but as the authors note, the majority of the substrate density is found interacting with the C terminal tails. This begs the question, why weren't multiple protein substrate bound forms observed (positioning of substrate within the chaperonin cavity). Why weren't multiple populations observed as seen previously by the Saibil group? Were their results flawed in some manner? It is interesting that a large amount of potential complexes had to be thrown out in the EM structural analysis. Could this thrown out population actually represent a heterogeneous GroEL-PePQ population? The authors can address this in the discussion section. This does not detract from the major conclusions of this paper that the substrate interacts with the C terminal tails manifesting a structural difference in the trapped forms (Does this state exist in vivo - no nucleotide), and that this interaction seems to impart a kinetic advantage.

C) The authors clarified their statements to emphasize the kinetic partitioning aspects of the chaperonin mechanism.

D) This reviewer appreciates the inclusion of this question in the discussion section because in vivo, individual chaperones don't fold alone. (emerging chain from ribosome interacts with nascent chaperones (trigger factor etc) and hsp70s.

E) This reviewer thanks the authors for expanding this explanation and interesting dangling point. An interesting corollary here is that the Hsp90 system has to unfold inactive kinases that are trapped in long lived metastable states which are also monomers. Reference to Agard's work with the Hsp90/Cdc37/Cdk4 complex may be appropriate here. This reviewer would suspect that addition of GroEL to the spontaneously folded states would not result in higher activity because once the trapped metastable species are formed, GroEL will be unable to rescue it (Unless another

chaperone system unfolds the metastable state, allowing the GroEL to engage with the partially folded states once again).

This reviewer has read the other reviews and by in large agrees with the authors comments and responses. one additional note. References for commitment rate measurements should perhaps include those that were measured the dodecamer glutamine synthetase, a monomer that needs to be partially unfolded in order to insure proper domain swapping events for oligomer formation. Here the activity measurements only reflects the readout of the formation of the downstream assembly but showed that the commitment rates were rapid compared with the simply following folding by activity measurements.

Response to Reviewer Comments

Reviewer 3:

Q.1 *minor point - C termini involved in pulling. Removal of pulling is agreed upon. However, there is still a matter of small disagreement with respect to an active vs passive mechanism. They make the argument using the hsp70 capture and prevention of back diffusion for a brownian ratchet type mechanism. This is not quite the same mechanism that is used GroEL because the Rapoport reference transport system is dependent on a leading strand protein unfolding from a threading perspective rather than a capture of an entire partially folded protein. In that there may be some further annealing of GGM repeats interacting with the partially unfolded protein, the actions may still be passive because the initial capture (for example initially at the interior rim) is an dynamic equilibrium process where simultaneous interaction of the substrate with the rim and end C termini GGM does not result in kinetically trapped complex at the top of the rim but rather, the initially bound substrate (perhaps represented by the state of the C terminal depleted species - may actually represent the initial trapped form)) has the opportunity to dissociate from the rim region (momentarily) and then further partition with or interacts with the tighter binding GGM tails.*

A.1 To a large extent, the argument presented by the reviewer is grounded in an issue of terminology around the words “active” versus “passive.” We have tried to be explicit in how we use these terms in the manuscript. We have outlined this usage in the Introduction and referenced there a more complete review of these issues (Z. Lin, H. Rye, Crit Rev Biochem Mol Bio. 41 (2006) 211–239). The sense in which we employ the term “active” with respect to GroEL folding mechanisms is specific: we distinguish between simple, infinite dilution (“passive”) models where the overall energetics of the isolated polypeptide are completely independent of the presence of GroEL. In a very real sense, even this class of model is not “fully” passive, as the GroEL machine must still hydrolyze ATP and proceed through cycles of substrate protein binding and release. Nonetheless, we prefer the use of the term “passive” in referring, in a simple way, to this class of model because it captures the essence of a model in which the only thing GroEL does is eliminate a side reaction (aggregation) through infinite dilution of the protein without doing anything to the individual folding intermediate’s conformational distribution or energetics. In other words, in a purely passive model, the intrinsic thermodynamic drive of the protein, and all the associated barrier crossing for the *monomeric* folding intermediate is assumed to be exactly the same in the presence and absence of GroEL. An “active” model, by contrast, assumes that the interaction with GroEL directly pumps energy into the system at the level of the individual polypeptide chain, either through unfolding or landscape smoothing or both (Z. Lin, H. Rye, Crit Rev Biochem Mol Bio. 41 (2006) 211–239). An alternate usage could be proposed, whereupon any action on the substrate protein that does not involve a directed, mechanical pulling or pushing force definitionally requires a “passive” model of GroEL function. However, as we argued in our previous response, this is not strictly correct. A purely binding-driven process like a Brownian ratchet can quite “actively” alter protein conformation or move objects or transport material across a membrane. From an energetic point of view, whether the protein moves through a pore, surface denatures on a piece of glass or is stretched across multiple binding sites in a chaperonin ring, does not really matter. The conformational distribution of the protein is directly altered as a result of one or more simple (i.e. passive) binding interactions with another entity, even if no directed mechanical pulling or pushing is

applied. If metabolic energy is then used to reverse these interaction (as it must be for a functional molecular chaperone), the sum total consequence of the binding/release cycle is, in point of fact, an active process of conformational alteration of the substrate protein. This is fundamentally distinct from the concept of infinite dilution, in which the conformational distribution of the monomeric protein is never altered by the presence of the chaperonin. As a final point, a force model of annealing onto a surface as a “purely” passive process would need to be clearly defined before it can be convincingly used as a counter-example to an active model.

Q.2 *Based on H. Saibils work, there certainly may be more intermediate GroEL complexes that have not been adequately resolved (i.e some substrate on the top, but as the authors note, the majority of the substrate density is found interacting with the C terminal tails. This begs the question, why weren't multiple protein substrate bound forms observed (positioning of substrate within the chaperonin cavity). Why weren't multiple populations observed as seen previously by the Saibil group? Were their results flawed in some manner? It is interesting that a large amount of potential complexes had to be thrown out in the EM structural analysis. Could this thrown out population actually represent a heterogenous GroEL-PepQ population? The authors can address this in the discussion section. This does not detract from the major conclusions of this paper that the substrate interacts with the C terminal tails manifesting a structural difference in the trapped forms (Does this state exist in vivo - no nucleotide), and that this interaction seems to impart a kinetic advantage.*

A.2 We have now added a new paragraph to the Discussion section placing the prior observations from the Saibil lab on MDH in context with the observations we present here with PepQ.

Some additional points:

(1) The PepQ folding intermediate is conformationally heterogenous on both the wild type GroEL and $\Delta 526$ rings. The much weaker observed density of the PepQ monomer on a wild type GroEL ring, however, indicates that this conformational heterogeneity is significantly greater for this complex. Nonetheless, the majority of the PepQ density is clearly located deep within the GroEL cavity when the protein is bound to a wild type GroEL ring, while it is shifted to a much higher position, toward the cavity rim, in the $\Delta 526$ complex.

(2) It is certainly possible that other sub-populations of bound PepQ exist, perhaps located at even more elevated positions toward the outer edge of the GroEL ring. Such a binding mode would be analogous to one sub-population of the non-native MDH monomer observed in work by Saibil and colleagues. However, if such a bound state exists for PepQ, it is so poorly populated that we could not detect it in our single-particle cryo-EM experiments.

(3) The particles that were rejected as part of the data processing pipeline outlined in the manuscript were eliminated because the conformation of the GroEL tetradecamer appeared badly distorted. It is possible that this sub-population of distorted oligomers contains externally or more unusually bound PepQ monomers. However, it is also possible that this sub-population simply consists of oligomers that are damaged or distorted for other trivial reasons of sample preparation. Because we cannot readily distinguish between these possibilities, we have

chosen to focus on the majority of particles that retain the well-established organization and overall symmetry of the GroEL tetradecamer. Within this population of well-structured tetradecamers, the only Pep-Q bound states observed are the ones presented.

(4) One important difference between the studies done by the Saibil group on MDH and our studies of PepQ has to do with the size of the protein. MDH has a molecular mass of 33 kDa while PepQ has a mass 52 kDa. The size of the protein, as well as its ability to interact with multiple apical domains, in addition to the C-termini, could easily have a profound impact on both the average position of the folding intermediate in the cavity, as well as the number and population of distinctly bound sub-populations. Additionally, there is no reason that the internal dynamics and conformational properties of any two folding intermediates are necessarily exactly the same. Thus detailed differences in the populations distributions of the non-native states of different proteins are not particularly surprising.

Q.3 *This reviewer thanks the authors for expanding this explanation and interesting dangling point. An interesting corollary here is that the Hsp90 system has to unfold inactive kinases that are trapped in long lived metastable states which are also monomers. Reference to Agards work with the Hsp90/Cdc37/Cdk4 complex may be appropriate here. This reviewer would suspect that addition of GroEL to the spontaneously folded states would not result in higher activity because once the trapped metastable species are formed, GroEL will be unable to rescue it (Unless another chaperone system unfolds the metastable state, allowing the GroEL to engage with the partially folded states once again).*

A.3 We certainly agree with the general point made here by the reviewer, that metastable intermediates are likely to be a common feature of the folding landscape of many essential cellular proteins, and that molecular chaperones may well target and unfold these states as a common feature of stimulated folding. However, the fundamental mechanisms of the Hsp90s and Hsp60s are quite distinct. While we are also intrigued by the parallels observed in the work of Agard's group and others on the Hsp90s and Hsp70s, we feel that a general discussion of this topic is well outside the scope of the current manuscript and would be far better addressed in a more general review of molecular chaperone action.

Q.4 *This reviewer has read the other reviews and by in large agrees with the authors comments and responses. one additional note. References for commitment rate measurements should perhaps include those that were measured the dodecamer glutamine synthetase, a monomer that needs to be partially unfolded in order to insure proper domain swapping events for oligomer formation. Here the activity measurements only reflects the readout of the formation of the downstream assembly but showed that the commitment rates were rapid compared with the simply following folding by activity measurements.*

A.4 We have now added a reference to the work done in the Fisher group on glutamine synthase folding by GroELS.